# Optimizing Posterior Samples for Bayesian Optimization via Rootfinding

**Taiwo A. Adebiyi**
University of Houston
taadebi2@uh.edu

**Bach Do**
University of Houston
bdo3@uh.edu

**Ruda Zhang**
University of Houston
rudaz@uh.edu

## Abstract

Bayesian optimization devolves the global optimization of a costly objective function to the global optimization of a sequence of acquisition functions. This inner-loop optimization can be catastrophically difficult if it involves posterior sample paths, especially in higher dimensions. We introduce an efficient global optimization strategy for posterior samples based on global rootfinding. It provides gradient-based optimizers with two sets of judiciously selected starting points, designed to combine exploration and exploitation. The number of starting points can be kept small without sacrificing optimization quality. Remarkably, even with just one point from each set, the global optimum is discovered most of the time. The algorithm scales practically linearly to high dimensions, breaking the curse of dimensionality. For Gaussian process Thompson sampling (GP-TS), we demonstrate remarkable improvement in both inner- and outer-loop optimization, surprisingly outperforming alternatives like EI and GP-UCB in most cases. Our approach also improves the performance of other posterior sample-based acquisition functions, such as variants of entropy search. Furthermore, we propose a sample-average formulation of GP-TS, which has a parameter to explicitly control exploitation and can be computed at the cost of one posterior sample.

## 1 Introduction

Bayesian optimization (BO) is a highly successful approach to the global optimization of expensive-to-evaluate black-box functions, with applications ranging from hyper-parameter training of machine learning models to scientific discovery and engineering design (Jones et al., 1998; Snoek et al., 2012; Frazier, 2018; Garnett, 2023). Many BO strategies are also backed by strong theoretical guarantees on their convergence to the global optimum (Srinivas et al., 2010; Bull, 2011; Russo & Van Roy, 2014; Chowdhury & Gopalan, 2017).

Consider the global optimization problem $\min_{\mathbf{x} \in \mathcal{X}} f(\mathbf{x})$ where $\mathbf{x} \in \mathcal{X} \subset \mathbb{R}^d$ represents the vector of input variables and $f(\mathbf{x}) \in \mathbb{R}$ the objective function which can be evaluated at a significant cost, subject to observation noise. At its core, BO is a sequential optimization algorithm that uses a probabilistic model of the objective function to guide its evaluation decisions. Starting with a prior probabilistic model and some initial data, BO derives an acquisition function $\alpha(\mathbf{x})$ from the posterior model, which is much easier to evaluate than the objective function and often has easy-to-evaluate derivatives. The acquisition function is then optimized globally, using off-the-shelf optimizers, to provide a location to evaluate the objective function. This process is iterated until some predefined stopping criteria are met.

Effectively there are two nested iterations in BO: the outer-loop seeks to optimize the objective function $f(\mathbf{x})$, and the inner-loop seeks to optimize the acquisition function $\alpha(\mathbf{x})$ at each BO iteration. The premise of BO is that the inner-loop optimization can be solved accurately and efficiently, so that the outer-loop optimization proceeds informatively with a negligible added cost. In fact, the convergence guarantees of many BO strategies assume *exact* global optimization of the acquisition function. However, the efficient and accurate global optimization of acquisition functions is less trivial than it is often assumed to be (Wilson et al., 2018).

Acquisition functions are, in general, highly non-convex and have many local optima. In addition, many common acquisition functions are mostly flat surfaces with a few peaks (Rana et al., 2017),

which take up an overwhelmingly large portion of the domain as the input dimension grows. This creates a significant challenge for generic global optimization methods.

Some acquisition functions involve sample functions from the posterior model, which need to be optimized globally. Gaussian process Thompson sampling (GP-TS) (Chowdhury & Gopalan, 2017) uses posterior sample paths directly as random acquisition functions. In many information-theoretic acquisition functions such as entropy search (ES) (Hennig & Schuler, 2012), predictive entropy search (PES) (Hernández-Lobato et al., 2014), max-value entropy search (MES) (Wang & Jegelka, 2017), and joint entropy search (JES) (Tu et al., 2022; Hvarfner et al., 2022), multiple posterior samples are drawn and optimized to find their global optimum location and/or value. Such acquisition functions are celebrated for their nice properties in BO: TS has strong theoretical guarantees (Russo & Van Roy, 2014; 2016) and can be scaled to high dimensions (Mutny & Krause, 2018); information-theoretic acquisition functions are grounded in principles for optimal experimental design (MacKay, 2003); and both types can be easily parallelized in synchronous batches (Shah & Ghahramani, 2015; Hernández-Lobato et al., 2017; Kandasamy et al., 2018). However, posterior sample paths are much more complex than other acquisition functions, as they fluctuate throughout the design space, and are less smooth than the posterior mean and marginal variance. The latter are the basis of many acquisition functions, such as expected improvement (EI) (Jones et al., 1998), probability of improvement (PI) (Kushner, 1964), and upper confidence bound (GP-UCB) (Srinivas et al., 2010). As a consequence, posterior sample paths have many more local optima, and the number scales exponentially with the input dimension.

While there is a rich literature on prior probabilistic models and acquisition functions for BO, global optimization algorithms for acquisition functions have received little attention. One class of global optimization methods is derivative-free, such as the dividing rectangles (DIRECT) algorithm (Jones et al., 1993), covariance matrix adaptation evolution strategy (CMA-ES) algorithms (Hansen et al., 2003), and genetic algorithms (Mitchell, 1998). Gradient-based multistart optimization, on the other hand, is often seen as the best practice to reduce the risk of getting trapped in local minima (Kim & Choi, 2021), and enjoys the efficiency of being embarrassingly parallelizable. For posterior samples, their global optimization may use joint sampling on a finite set of points (Kandasamy et al., 2018), or approximate sampling of function realizations followed by gradient-based optimization (Hernández-Lobato et al., 2014; Mutny & Krause, 2018). The selection of starting points is crucial for the success of gradient-based multistart optimization. This selection can be deterministic (e.g., grid search), random (Bergstra & Bengio, 2012; Balandat et al., 2020), or adaptive (Feo & Resende, 1995).

In this paper, we propose an adaptive strategy for selecting starting points for gradient-based multistart optimization of posterior samples. This algorithm builds on the decomposition of posterior samples by pathwise conditioning, taps into robust software in univariate function computation based on univariate global rootfinding, and exploits the separability of multivariate GP priors. Our key contributions include:

- A novel strategy for the global optimization of posterior sample paths. The starting points are dependent on the posterior sample, so that each is close to a local optimum that is a candidate for the global optimum. The selection algorithm scales linearly to high dimensions.
- We give empirical results across a diverse set of problems with input dimensions ranging from 2 to 16, establishing the effectiveness of our optimization strategy. Although our algorithm is proposed for the inner-loop optimization of posterior samples, perhaps surprisingly, we see significant improvement in outer-loop optimization performance, which often allows acquisition functions based on posterior samples to converge faster than other common acquisition functions.
- A new acquisition function via the posterior sample average that explicitly controls the exploration–exploitation balance (Chapelle & Li, 2011), and can be generated at the same cost as one posterior sample.

## 2   GENERAL BACKGROUND

**Gaussian Processes.** Consider an unknown function $f_{\text{true}} : \mathcal{X} \mapsto \mathbb{R}$, where domain $\mathcal{X} \subset \mathbb{R}^d$. We can collect noisy observations of the function through the model $y^i = f_{\text{true}}(\mathbf{x}^i) + \varepsilon^i$, $i \in \{1, \cdots, n\}$, with $\boldsymbol{\varepsilon} \sim \mathcal{N}_n(0, \boldsymbol{\Sigma})$. To model the function $f_{\text{true}}$, we use a Gaussian process (GP) as the prior probabilistic model: $f \sim \pi \in \mathcal{GP}$. A GP is a random function $f$ such that for any finite set of points $X = \{\mathbf{x}^i\}_{i=1}^n$, $n \in \mathbb{N}$, the values $\mathbf{f}_n = (f(\mathbf{x}^i))_{i=1}^n$ have a multivariate Gaussian distribution

$\mathcal{N}_n(\boldsymbol{\mu}_n, \mathbf{K}_{n,n})$, with mean $\boldsymbol{\mu}_n = (\mu(\mathbf{x}^i))_{i=1}^n$ and covariance $\mathbf{K}_{n,n} = [\kappa(\mathbf{x}^i, \mathbf{x}^j)]_{i \in n}^{j \in n}$. Here, $\mu(\mathbf{x})$ is the mean function and $\kappa(\mathbf{x}, \mathbf{x}')$ is the covariance function.

**Decoupled Representation of GP Posteriors.** Given a dataset $\mathcal{D} = \{(\mathbf{x}^i, y^i)\}_{i=1}^n$, the posterior model $f|\mathcal{D}$ is also a GP. Samples from the posterior have a decoupled representation called pathwise conditioning, originally proposed in (Wilson et al., 2020; 2021):

$$(f|\mathcal{D})(\mathbf{x}) \stackrel{\mathrm{d}}{=} f(\mathbf{x}) + \boldsymbol{\kappa}_{\cdot,n}(\mathbf{x})(\mathbf{K}_{n,n} + \boldsymbol{\Sigma})^{-1}(\mathbf{y} - \mathbf{f}_n - \boldsymbol{\varepsilon}), \quad f \sim \pi, \ \boldsymbol{\varepsilon} \sim \mathcal{N}_n(0, \boldsymbol{\Sigma}), \quad (1)$$

where $f(\mathbf{x})$ is a sample path from the GP prior, $\boldsymbol{\kappa}_{\cdot,n}(\mathbf{x}) = (\kappa(\mathbf{x}, \mathbf{x}^i))_{i=1}^n$ is the canonical basis, $\mathbf{f}_n = (f(\mathbf{x}^i))_{i=1}^n$, and $\boldsymbol{\varepsilon}$ is a sample of the noise. We may interpret $\mathbf{f}_n + \boldsymbol{\varepsilon}$ as a sample from the prior distribution of the observations $\mathbf{y} = (y^i)_{i=1}^n$. This representation has its roots in Matheron's update rule that transforms a joint distribution of Gaussian variables into a conditional one (see e.g., Hoffman & Ribak (1991)). This formula is exact, in that $\stackrel{\mathrm{d}}{=}$ denotes equality in distribution, and it preserves the differentiability of the prior sample. It is also computationally efficient for posterior sampling: the cost is independent of the input dimension $d$, linear in the data size $n$ at evaluation time, and the weight vector for $\boldsymbol{\kappa}_{\cdot,n}(\mathbf{x})$ can be solved accurately using an iterative algorithm that scales linearly with $n$ (Lin et al., 2023).

# 3 GLOBAL OPTIMIZATION OF POSTERIOR SAMPLE PATHS

In this section, we introduce an efficient algorithm for the global optimization of posterior sample paths. For this, we exploit the separability of prior samples and useful properties of posterior samples to judiciously select a set of starting points for gradient-based multistart optimizers.

**Assumptions.** Following Section 2, we impose a few common assumptions throughout this paper: (1) the domain is a hypercube: $\mathcal{X} = \prod_{i=1}^d [\underline{x}_i, \overline{x}_i]$; (2) prior covariance is separable: $\kappa(\mathbf{x}, \mathbf{x}') = \prod_{i=1}^d \kappa_i(x_i, x_i')$; (3) prior samples are continuously differentiable: $f(\mathbf{x}; \omega) \in C^1$. Without loss of generality, we also assume that the prior mean $\mu(\mathbf{x}) = 0$: any non-zero mean function can be subtracted from the data by replacing $f_{\text{true}}$ with $f_{\text{true}} - \mu$. While additive and multiplicative compositions of univariate kernels can be used in the prior (Duvenaud et al., 2013), assumption (2) is the most popular choice in BO. It is possible to extend our method to generalized additive models.

## 3.1 TS-ROOTS ALGORITHM

We observe that, given the assumptions, a posterior sample in eq. (1) can be rewritten as:

$$\widetilde{f}(\mathbf{x}; \widetilde{\omega}) = f(\mathbf{x}; \omega) + b(\mathbf{x}; \widetilde{\omega}), \quad f(\mathbf{x}; \omega) \approx \prod_{i=1}^d f_i(x_i; \omega_i), \quad b(\mathbf{x}; \widetilde{\omega}) = \sum_{j=1}^n v_j \kappa(\mathbf{x}, \mathbf{x}^j). \quad (2)$$

Here, the prior sample $f(\mathbf{x})$ is approximated as a separable function determined by the random bits $\omega$ (see Appendix C.1). Data adjustment $b(\mathbf{x})$ is a sum of the canonical basis with coefficients $\mathbf{v} = (\mathbf{K}_{n,n} + \boldsymbol{\Sigma})^{-1}(\mathbf{y} - \mathbf{f}_n - \boldsymbol{\varepsilon})$. Both the data adjustment and the posterior sample are determined by the random bits $\widetilde{\omega} = (\omega, \boldsymbol{\varepsilon})$. In the following, we denote the prior and the posterior samples as $f_\omega$ and $\widetilde{f}_{\widetilde{\omega}}$, respectively. Our goal is to find the global minimum $(\widetilde{\mathbf{x}}_{\widetilde{\omega}}^\star, \widetilde{f}_{\widetilde{\omega}}^\star)$ of the posterior sample $\widetilde{f}_{\widetilde{\omega}}(\mathbf{x})$.

The global minimization of a generic function, in principle, requires finding all its local minima and then selecting the best among them. However, computationally efficient approaches to this problem are lacking even in low dimensions and, more pessimistically, the number of local minima grows exponentially as the domain dimension increases. The core idea of this work is to use the prior sample $f_\omega$ as a surrogate of the posterior sample $\widetilde{f}_{\widetilde{\omega}}$ for global optimization. Another key is to exploit the separability of the prior sample for efficient representation and ordering of its local extrema.

We define *TS-roots* as a global optimization algorithm for GP posterior samples (given the assumptions) via gradient-based multistart optimization. The starting points include: (1) a subset $S_e$ of the

---

**Algorithm 1** `TS-roots`: Optimizing posterior samples via rootfinding

---

**Input:** prior samples $f(\mathbf{x})$, $\boldsymbol{\varepsilon}$; prior covariances $\kappa(\mathbf{x}, \mathbf{x}')$, $\boldsymbol{\Sigma}$; dataset $\mathcal{D}$; set sizes $n_\mathrm{o}, n_\mathrm{e}, n_\mathrm{x}$.

1: $S_\mathrm{o} \leftarrow \mathtt{minsort}(f(\mathbf{x}), n_\mathrm{o})$      $\triangleright$ Smallest $n_\mathrm{o}$ local minima of the prior sample (Algorithm 4)
2: $[\widetilde{\mathbf{f}}_\mathrm{e}, I_\mathrm{e}] \leftarrow \mathtt{mink}(\widetilde{f}(S_\mathrm{o}), n_\mathrm{e})$      $\triangleright$ Smallest $n_\mathrm{e}$ values and indices of the posterior sample in $S_\mathrm{o}$
3: $[\widetilde{\mathbf{f}}_\mathrm{x}, I_\mathrm{x}] \leftarrow \mathtt{mink}(\overline{f}(X), n_\mathrm{x})$      $\triangleright$ Smallest $n_\mathrm{x}$ values and indices of the posterior mean in $X$
4: $S_\mathrm{e} \leftarrow S_\mathrm{o}[I_\mathrm{e}, :]$, $S_\mathrm{x} \leftarrow X[I_\mathrm{x}, :]$      $\triangleright$ Starting points: exploration set and exploitation set
5: $[\widetilde{\mathbf{x}}^\star, \widetilde{f}^\star] \leftarrow \mathtt{minimize}(\widetilde{f}(\mathbf{x}), S)$, $S = S_\mathrm{e} \cup S_\mathrm{x}$      $\triangleright$ Gradient-based multistart optimization
**Output:** Thompson sample $\widetilde{\mathbf{x}}^\star$      $\triangleright$ Global minimum of the posterior sample

---

local minima $\breve{X}_\omega$ of a corresponding prior sample; and (2) a subset $S_\mathrm{x}$ of the observed locations $X$. We call $S_\mathrm{e}$ the exploration set and $S_\mathrm{x}$ the exploitation set. Specifically, let

$$S_\mathrm{o} = \arg \mathop{\mathrm{mink}}_{\mathbf{x} \in \breve{X}_\omega}(f_\omega(\mathbf{x}), n_\mathrm{o}) \tag{3}$$

be the $n_\mathrm{o}$ smallest local minima of the prior sample. The set of starting points, $S$, is defined as:

$$S = S_\mathrm{e} \cup S_\mathrm{x}, \quad S_\mathrm{e} = \arg \mathop{\mathrm{mink}}_{\mathbf{x} \in S_\mathrm{o}}(\widetilde{f}_{\widetilde{\omega}}(\mathbf{x}), n_\mathrm{e}), \quad S_\mathrm{x} = \arg \mathop{\mathrm{mink}}_{\mathbf{x} \in X}(\widetilde{f}_{\widetilde{\omega}}(\mathbf{x}), n_\mathrm{x}). \tag{4}$$

Algorithm 1 outlines the procedure for TS-roots. Here, $n_\mathrm{e}$ and $n_\mathrm{x}$ are imposed to control the cost of gradient-based multistart optimization, and $n_\mathrm{o}$ is set to limit the number of evaluations of $\widetilde{f}_{\widetilde{\omega}}$ in the determination of $S_\mathrm{e}$. Considering the cost difference, we can have $n_\mathrm{o} \gg n_\mathrm{e}$. We observe that $n_\mathrm{e}$ and $n_\mathrm{x}$ can be set to small values, and $n_\mathrm{o}$ to a moderate value, without sacrificing the quality of optimization, see Appendix D. The TS-roots algorithm scales linearly in $d$, see Appendix C.5.

## 3.2 RELATIONS BETWEEN THE LOCAL MINIMA OF PRIOR AND POSTERIOR SAMPLES

Figure 1 shows several posterior samples $\widetilde{f}_{\widetilde{\omega}}$ in one and two dimensions, each marked with its local minima $\breve{X}_{\widetilde{\omega}}$ and global minimum $\widetilde{\mathbf{x}}^\star_{\widetilde{\omega}}$. Here the exploration set $S_\mathrm{e}$ is the local minima $\breve{X}_\omega$ of $f_\omega$, and the exploitation set $S_\mathrm{x}$ is the observed locations $X$. We make the following observations:

(1) The prior sample $f_\omega$ is more complex than the data adjustment $b$ in the sense that it is less smooth and has more critical points. The comparison of smoothness can be made rigorous in various ways: for example, for GPs with a Matérn covariance function where the smoothness parameter is finite, $f_\omega$ is almost everywhere one time less differentiable than $b$ (see e.g., Garnett (2023) Sec. 10.2, Kanagawa et al. (2018)).

(2) Item (1) implies that when the prior sample $f_\omega$ is added to the smoother landscape of $b$, each local minimum $\breve{\mathbf{x}}_\omega$ of $f_\omega$ will be located near a local minimum $\breve{\mathbf{x}}_{\widetilde{\omega}}$ of the posterior sample $\widetilde{f}_{\widetilde{\omega}}$. Away from the observed locations $X$, each $\breve{\mathbf{x}}_{\widetilde{\omega}}$ is closely associated with an $\breve{\mathbf{x}}_\omega$, with minimal change in location. In the vicinity of $X$, an $\breve{\mathbf{x}}_{\widetilde{\omega}}$ may have both a data point $\mathbf{x}^i$ and an $\breve{\mathbf{x}}_\omega$ nearby, but because of the smoothness difference of $f_\omega$ and $b$, in most cases the one closest to $\breve{\mathbf{x}}_{\widetilde{\omega}}$ is an $\breve{\mathbf{x}}_\omega$.

(3) It is possible that near $X$, sharp changes in $f_\omega$ may require sharp adjustments to the data, which may move some $\breve{\mathbf{x}}_\omega$ by a significant distance, or create new $\breve{\mathbf{x}}_{\widetilde{\omega}}$ that are not near any $\breve{\mathbf{x}}_\omega$ or any $\mathbf{x}^i$.

(4) Searching from $\mathbf{x}^i$ with good observed values can discover good $\breve{\mathbf{x}}_{\widetilde{\omega}}$ in the vicinity of $X$, which can pick up some local minima not readily discovered by $\breve{X}_\omega$. This is especially true if $f_\omega$ is relatively flat near $\mathbf{x}^i$.

(5) Since the posterior sample $\widetilde{f}_{\widetilde{\omega}}$ adapts to the dataset, searches from $\mathbf{x}^i$ will tend to converge to a good $\breve{\mathbf{x}}_{\widetilde{\omega}}$ among all the local optima near $\mathbf{x}^i$. Even if the searches from $X$ cannot discover all the local minima in its vicinity, they tend to discover a good subset of them. Therefore, (4) can help address the issue in (3), if not fully eliminating it. By combining subsets of $\breve{X}_\omega$ and $X$, we can expect that the set of local minima of $\widetilde{f}_{\widetilde{\omega}}$ discovered with these starting points includes the global minimum $\widetilde{\mathbf{x}}^\star_{\widetilde{\omega}}$ with a high probability with respect to $\widetilde{\omega}$.

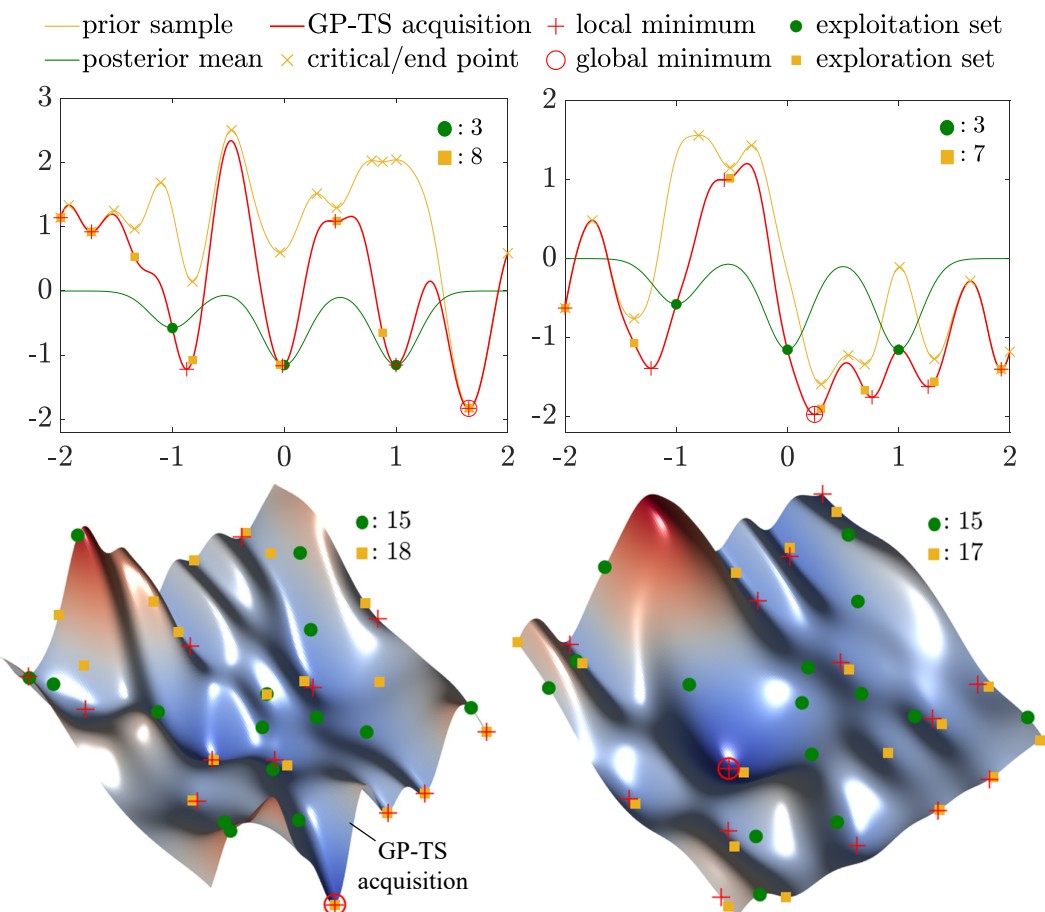

Figure 1: Illustrations of exploration and exploitation sets for the global optimization of GP-TS acquisition functions in one dimension (top row) and two dimensions (bottom row). *Left column:* When the global minimum $\widetilde{\mathbf{x}}_{\widetilde{\omega}}^{\star}$ of the GP-TS acquisition function lies outside the interpolation region, it is typically identified by starting the gradient-based multistart optimizer at a local minimum of the prior sample. *Right column:* When $\widetilde{\mathbf{x}}_{\widetilde{\omega}}^{\star}$ is within the interpolation region, it can be found by starting the optimizer at either an observed location or a local minimum of the prior sample.

### 3.3 A REPRESENTATION OF PRIOR SAMPLE LOCAL MINIMA

For each component function $f_i(x_i; \omega_i)$ of the prior sample $f_\omega(\mathbf{x})$, define $h_i(\underline{x}_i) = f_i'(\underline{x}_i)$, $h_i(\overline{x}_i) = -f_i'(\overline{x}_i)$, and $h_i(x_i) = f_i''(x_i)$ for $x_i \in (\underline{x}_i, \overline{x}_i)$. We call a coordinate $\xi_i \in [\underline{x}_i, \overline{x}_i]$ of *mono type* if $f_i(\xi_i)h_i(\xi_i) > 0$ and call it of *mixed type* if $f_i(\xi_i)h_i(\xi_i) < 0$. Let $\dot{\Xi}_i = \{\xi_{i,j}\}_{j=1}^{r_i}$ be the set of interior critical points of $f_i$ such that $\xi_{i,j} \in (\underline{x}_i, \overline{x}_i)$ and $f_i'(\xi_{i,j}) = 0$, $j \in \{1, \cdots, r_i\}$. Denote $\xi_{i,0} = \underline{x}_i$ and $\xi_{i,r_i+1} = \overline{x}_i$. Partition the set of candidate coordinates $\Xi_i = \{\xi_{i,j}\}_{j=0}^{r_i+1}$ into mono type $\Xi_i^{(0)}$ and mixed type $\Xi_i^{(1)}$. Proposition 1 gives a representation of the sets of strong local extrema of the prior sample. Its proof and the set sizes therein are given in Appendix A.

**Proposition 1** *The set of strong local minima of the prior sample $f_\omega(\mathbf{x})$ can be written as:*

$$\breve{X}_\omega = \breve{X}_\omega^- \sqcup \breve{X}_\omega^+, \quad \breve{X}_\omega^- = \{\boldsymbol{\xi} \in \Xi^{(1)} : f_\omega(\boldsymbol{\xi}) < 0\}, \quad \breve{X}_\omega^+ = \{\boldsymbol{\xi} \in \Xi^{(0)} : f_\omega(\boldsymbol{\xi}) > 0\}, \quad (5)$$

*where tensor grids $\Xi^{(j)} = \prod_{i=1}^d \Xi_i^{(j)}$, $j \in \{0, 1\}$. The set $\widehat{X}_\omega$ of strong local maxima of $f_\omega(\mathbf{x})$ has an analogous representation, and satisfies $\widehat{X}_\omega \sqcup \breve{X}_\omega = \Xi^{(0)} \sqcup \Xi^{(1)}$, where $\sqcup$ is the disjoint union.*

**Critical Points of Univariate Functions via Global Rootfinding.** To compute the set $\mathring{\Xi}_i$ of all critical points of $f_i$ is to compute all the roots of the derivative $f_i'$ on the interval $[\underline{x}_i, \overline{x}_i]$. Since

$f'_i$ is continuous, this can be solved robustly and efficiently by approximating the function with a Chebyshev or Legendre polynomial and solving a structured eigenvalue problem (see e.g., Trefethen (2019)). The `roots` algorithm for global rootfinding based on polynomial approximation is given as Algorithm 3 in Appendix C.

### 3.4 Ordering of Prior Sample Local Minima

While the size of $\breve{X}_\omega$ grows exponentially in domain dimension $d$, its representation in eq. (5) enables an efficient algorithm to compute the best subset $S_o$ (eq. (3)) without enumerating its elements.

With eq. (5), we see that $\breve{X}_\omega^-$ consists of all the local minima of $f_\omega$ with negative function values. Consider the case where $\breve{X}_\omega^-$ has at least $n_o$ elements so that in the definition of $S_o$ we can replace $\breve{X}_\omega$ with $\breve{X}_\omega^-$, which in turn can be replaced with $\Xi^{(1)}$. As we will show later, the problem of finding the largest elements of $|f_\omega(\mathbf{x})|$ within $\Xi^{(1)}$ is easier than finding the smallest negative elements of $f_\omega(\mathbf{x})$. Once the former is solved, we can solve the latter simply by removing the positive elements. Therefore, we convert the problem of eq. (3) into three steps:

1. $S^{(1)} = \arg\max_{\mathbf{x} \in \Xi^{(1)}}(|f_\omega(\mathbf{x})|, \alpha n_o)$, with buffer coefficient $\alpha \geq 1$;

2. $\breve{S}^- = \{\mathbf{x} \in S^{(1)} : f_\omega(\mathbf{x}) < 0\}$, so that $\breve{S}^- \subseteq \breve{X}_\omega^-$;

3. $S_o = \arg\min_{\mathbf{x} \in \breve{S}^-}(f_\omega(\mathbf{x}), n_o)$, assuming that $|\breve{S}^-| \geq n_o$.

The last two steps are by enumeration and straightforward. The first step can be solved efficiently using min-heaps, with a time complexity that scales linearly in $\sum_{i=1}^d |\Xi_i^{(1)}|$ rather than $\prod_{i=1}^d |\Xi_i^{(1)}|$, see Appendix B. The coefficient $\alpha$ is chosen so that $|\breve{S}^-| \geq n_o$. The case when $|\breve{X}_\omega^-| < n_o < |\breve{X}_\omega|$ can be handled similarly. If $n_o \geq |\breve{X}_\omega|$, no subsetting is needed. The overall procedure to compute $S_o$ is given in Algorithm 4 in Appendix C.

## 4 Sample-average Posterior Function

We finally propose a sample-average posterior function that explicitly controls the exploration–exploitation balance and, notably, can be generated at the cost of generating one posterior sample. Let $\widetilde{\mu}(\mathbf{x}) = \boldsymbol{\kappa}_{\cdot,n}(\mathbf{x})(\mathbf{K}_{n,n} + \boldsymbol{\Sigma})^{-1}\mathbf{y}$ be the posterior mean function. For noiseless observations with $\widetilde{\omega} = \omega$, we can rewrite the GP posterior function in eq. (2) as $\widetilde{f}_\omega(\mathbf{x}) = f_\omega(\mathbf{x}) + \widetilde{\mu}(\mathbf{x}) + \xi_\omega(\mathbf{x})$, where $\xi_\omega(\mathbf{x}) = -\boldsymbol{\kappa}_{\cdot,n}(\mathbf{x})\mathbf{K}_{n,n}^{-1}\mathbf{f}_n$. Define $\alpha_{\text{aTS}}(\mathbf{x}) = \frac{1}{N_c}\sum_{j=1}^{N_c} \widetilde{f}_\omega^j(\mathbf{x})$ as the sample-average posterior function, where $\widetilde{f}_\omega^j(\mathbf{x})$ are samples generated from the GP posterior and $N_c \in \mathbb{N}_{>0}$. Since $\widetilde{\mu}(\mathbf{x})$ is deterministic, and the scaled prior sample $\frac{1}{\sqrt{N_c}}f_\omega^j(\mathbf{x})$ can be written as $\frac{1}{\sqrt{N_c}}f_\omega^j(\mathbf{x}) \stackrel{\text{iid}}{\sim} \mathcal{GP}(0, \frac{1}{N_c}\kappa(\mathbf{x},\mathbf{x}'))$, we have $\alpha_{\text{aTS}}(\mathbf{x}) = \mu(\mathbf{x}) + \frac{1}{\sqrt{N_c}}(f_\omega(\mathbf{x}) + \xi_\omega(\mathbf{x}))$, where the first and second terms favor exploitation and exploration, respectively. Thus, we can consider $N_c$ as an exploration–exploitation control parameter that, at large values, prioritizes exploitation by concentrating the conditional distribution of the global minimum location, i.e., $p(\mathbf{x}^\star|\mathcal{D})$, at the minimum location of $\widetilde{\mu}(\mathbf{x})$, see Figure 13 in Appendix I. With $\alpha_{\text{aTS}}(\mathbf{x})$, we can reproduce $\widetilde{f}_\omega(\mathbf{x})$ and the GP mean function $\widetilde{\mu}(\mathbf{x})$ by setting $N_c = 1$ and $N_c = \infty$, respectively.

## 5 Related Works

**Sampling from Gaussian Process Posteriors.** A prevalent method to sample GP posteriors with stationary covariance functions is via weight-space approximations based on Bayesian linear models of random Fourier features (Rahimi & Recht, 2007). This method, unfortunately, is subject to the variance starvation problem (Mutny & Krause, 2018; Wilson et al., 2020) which can be mitigated using more accurate feature representations (see e.g., Hensman et al. (2018); Solin & Särkkä (2020)). An alternative is pathwise conditioning (Wilson et al., 2020) that draws GP posterior samples by updating the corresponding prior samples. The decoupled representation of the pathwise conditioning can be further reformulated as two stochastic optimization problems for the posterior

mean and an uncertainty reduction term, which are then efficiently solved using stochastic gradient descent (Lin et al., 2023).

**Optimization of Acquisition Functions.** While their global optima guarantee the Bayes' decision rule, BO acquisition functions are highly non-convex and difficult to optimize (Wilson et al., 2018). Nevertheless, less attention has been given to the development of robust algorithms for optimizing these acquisition functions. For this inner-loop optimization, gradient-based optimizers are often selected because of their fast convergence and robust performance (Daulton et al., 2020). The implementation of such optimizers is facilitated by Monte Carlo (MC) acquisition functions whose derivatives are easy to evaluate (Wilson et al., 2018). Gradient-based optimizers also allow multistart settings that use a set of starting points which can be, for example, midpoints of data points (Jones, 2001), uniformly distributed samples over the input variable space (Frazier, 2018; Ament et al., 2023), or random points from a Latin hypercube design (Wang et al., 2020). However, multistart-based methods with random search may have difficulty determining the non-flat regions of acquisition functions, especially in high dimensions (Rana et al., 2017). The log reformulation approach is a good solution to the numerical pathology of flat acquisition surface over large regions of the input variable space (Ament et al., 2023). While this approach works for acquisition functions prone to the flat surface issue such as the family of EI-based acquisition functions, its performance has yet to be evaluated for acquisition functions with many local minima like those based on posterior samples.

**Posterior Sample-Based Acquisition Functions.** As discussed in Section 1, the family of posterior sample-based acquisition functions is determined from samples of the posterior. GP-TS (Chowdhury & Gopalan, 2017) is a notable member that extends the classical TS for finite-armed bandits to continuous settings of BO (see algorithms in Appendix E). GP-TS prefers exploration by the mechanism that iteratively samples a function from the GP posterior of the objective function, optimizes this function, and selects the resulting solution as the next candidate for objective evaluation. To further improve the exploitation of GP-TS, the sample mean of MC acquisition functions can be defined from multiple samples of the posterior (Wilson et al., 2018; Balandat et al., 2020). Different types of MC acquisition functions can also be used to inject beliefs about functions into the prior (Hvarfner et al., 2024).

## 6    RESULTS

We assess the performance of TS-roots in optimizing benchmark functions. We then compare the quality of solutions to the inner-loop optimization of GP-TS acquisition functions obtained from our proposed method, a gradient-based multistart optimizer with uniformly random starting points, and a genetic algorithm. We also show how TS-roots can improve the performance of MES. Finally, we propose a new sample-average posterior function and show how it affects the performance of GP-TS. The experimental details for the presented results are in Appendix H. Our implementation of the TS-roots algorithm is available at `https://github.com/UQUH/TSRoots`.

**Optimizing Benchmark Functions.** We test the empirical performance of TS-roots on challenging minimization problems of five benchmark functions: the 2D Schwefel, 4D Rosenbrock, 10D Levy, 16D Ackley, and 16D Powell functions (Surjanovic & Bingham, 2013). The analytical expressions for these functions and their global minimum are given in Appendix F.

In each optimization iteration, we record the best observed value of the error $\log(y_{\min} - f^\star)$ and the distance $\log(\|\mathbf{x}_{\min} - \mathbf{x}^\star\|)$, where $y_{\min}$, $\mathbf{x}_{\min}$, $f^\star$, and $\mathbf{x}^\star$ are the best observation of the objective function in each iteration, the corresponding location of the observation, the true minimum value of the objective function, and the true minimum location, respectively. We compare the optimization results obtained from TS-roots and other BO methods, including GP-TS using decoupling sampling with random Fourier features (TS-DSRF), GP-TS with random Fourier features (TS-RF), expected improvement (EI) (Jones et al., 1998), lower confidence bound (LCB)—the version of GP-UCB (Srinivas et al., 2010) for minimization problems, and LogEI (Ament et al., 2023).

Figure 2 shows the medians and interquartile ranges of solution values obtained from 20 runs of each of the considered BO methods. The corresponding histories of solution locations are in Figure 10 of Appendix I. With a fair comparison of outer-loop results (detailed in Appendix H), TS-roots surprisingly shows the best performance on the 2D Schwefel, 16D Ackley, and 16D Powell functions, and gives competitive results in the 4D Rosenbrock and 10D Levy problems. Notably, TS-roots

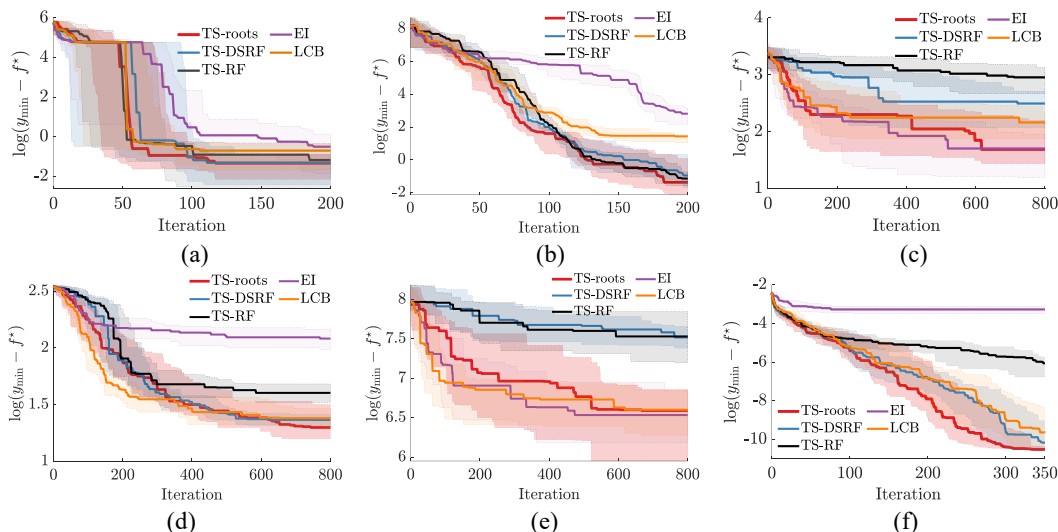

Figure 2: Outer-loop optimization results for the (a) 2D Schwefel function, (b) 4D Rosenbrock function, (c) 10D Levy function, (d) 16D Ackley function, (e) 16D Powell function, and (f) ten-bar truss problem. The plots are histories of medians and interquartile ranges of solution values from 20 runs of TS-roots, TS-DSRF (i.e., TS using decoupled sampling with random Fourier features), TS-RF (i.e., TS using random Fourier features), EI, and LCB.

recommends better solutions than its counterparts, TS-DSRF and TS-RF, in high-dimensional problems and offers competitive performance in low-dimensional problems. Across all the examples, EI and LCB tend to perform better in the initial stages, while TS-roots shows fast improvement in later stages. This is because GP-TS favors exploration, which delays rewards. The exploration phase, in general, takes longer for higher-dimensional problems. TS-roots outperforms LogEI on the 2D Schwefel, 4D Rosenbrock, 10D Levy, and 16D Ackley functions (detailed in Figure 9 of Appendix I).

**Optimizing Real-world Problem.** We implement TS-roots to optimize an engineered ten-bar truss structure (see Appendix G). Ten design variables of the truss are the cross-sectional areas of the truss members. The objective is to minimize a weighted sum of the scaled total cross-sectional area and the scaled vertical displacement at a node of interest.

Figures 2(f) and 9(f) show the outer-loop optimization results for the truss obtained from 20 runs of each BO method, where $f^\star$ is a lower bound of the best objective function value we observed from all runs. TS-roots provides the best optimization result with rapid convergence.

**Optimizing GP-TS Acquisition Functions via Rootfinding.** We assess the quality of solutions and computational cost for the inner-loop optimization of GP-TS acquisition functions by the proposed global optimization algorithm, referred to as rootfinding hereafter. We do so by computing the optimized values $\alpha_k^\star$ of the GP-TS acquisition functions, the corresponding solution points $\mathbf{x}_k^\star$, and the CPU times $t_k$ required for optimizing the acquisition functions during the optimization process. For low-dimensional problems of the 2D Schwefel and 4D Rosenbrock functions, we also compute the exact global solution points $\mathbf{x}_k^t$ of the GP-TS acquisition functions by starting the gradient-based optimizer at a large number of initial points (set as $10^4$), which is much larger than the maximum number of starting points set for TS-roots. For comparison, we extend the same GP-TS acquisition functions to inner-loop optimization using a gradient-based multistart optimizer with random starting points (i.e., random multistart) and a genetic algorithm. In each outer-loop optimization iteration, the number of starting points for the random multistart and the population size of the genetic algorithm are equal to the number of starting points recommended for rootfinding. The same termination conditions are used for the three algorithms.

Figure 3 shows the comparative performance of the inner-loop optimization for low-dimensional cases: the 2D Schwefel and 4D Rosenbrock functions. We see that the optimized acquisition func-

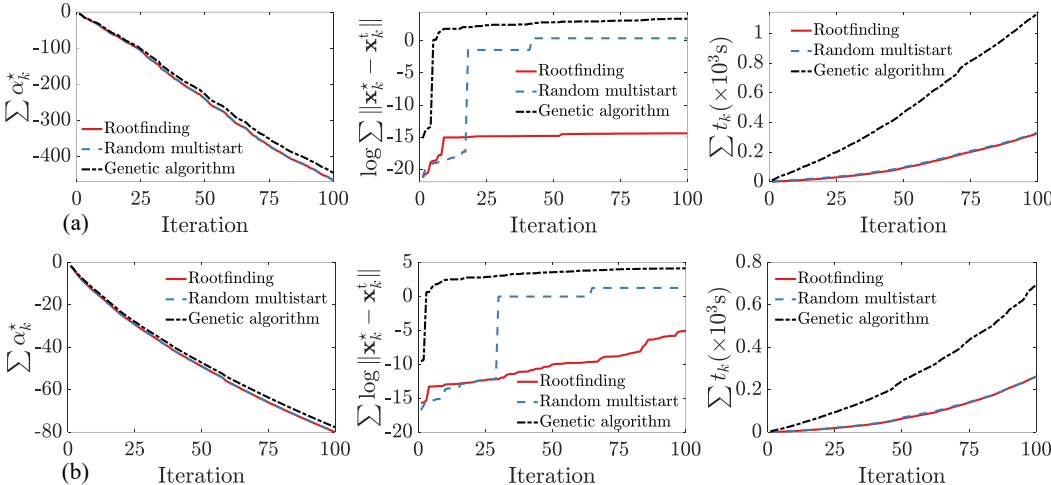

Figure 3: Inner-loop optimization results by rootfinding, a gradient-based multistart optimizer with random starting points (random multistart), and a genetic algorithm for (a) the 2D Schwefel and (b) 4D Rosenbrock functions. The plots are cumulative values of optimized GP-TS acquisition functions $\alpha_k^\star$, cumulative distances between new solution points $\mathbf{x}_k^\star$ and the true global minima $\mathbf{x}_k^t$ of the acquisition functions, and cumulative CPU times $t_k$ for optimizing the acquisition functions.

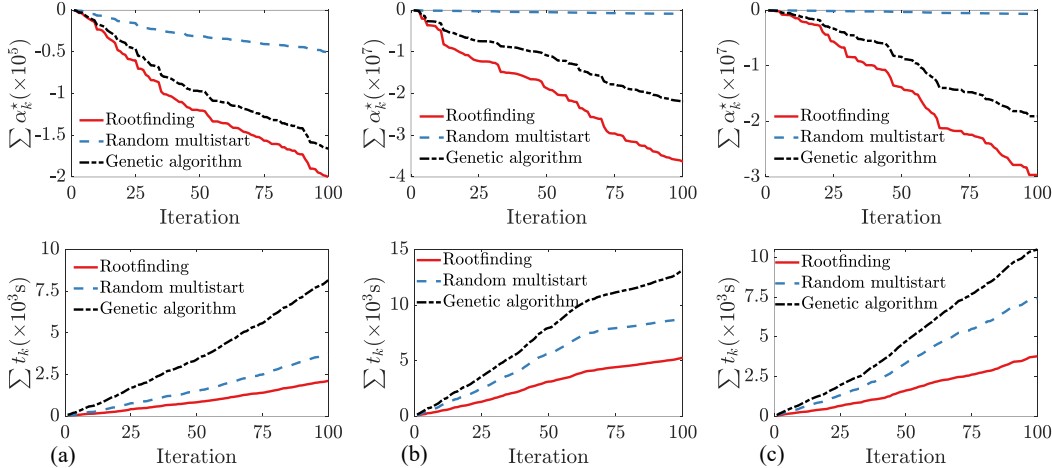

Figure 4: Inner-loop optimization results by rootfinding, a gradient-based multistart optimizer with random starting points (random multistart), and a genetic algorithm for (a) the 10D Levy, (b) 16D Ackley, and (c) 16D Powell functions. The plots are cumulative values of optimized GP-TS acquisition functions $\alpha_k^\star$ and cumulative CPU times $t_k$ for optimizing the acquisition functions.

tion values and the optimization runtimes by rootfinding and the random multistart algorithm are almost identical, both of which are much better than those by the genetic algorithm. Rootfinding gives the best quality of the new solution points in both cases, while the genetic algorithm gives the worst. Notably in higher-dimensional settings of the 10D Levy, 16D Ackley, and 16D Powell functions shown in Figure 4, rootfinding performs much better than the random multistart and genetic algorithm in terms of optimized acquisition values and optimization runtimes, which verifies the importance of the judicious selection of starting points for global optimization of the GP-TS acquisition functions and the efficiency of rootfinding in high dimensions. The performance of random multistart becomes worse in higher dimensions. Appendix I provides additional results for gradient-based multistart optimization using two other initialization schemes: uniform grid and Latin hypercube sampling. Rootfinding outperforms both, especially in higher dimensions.

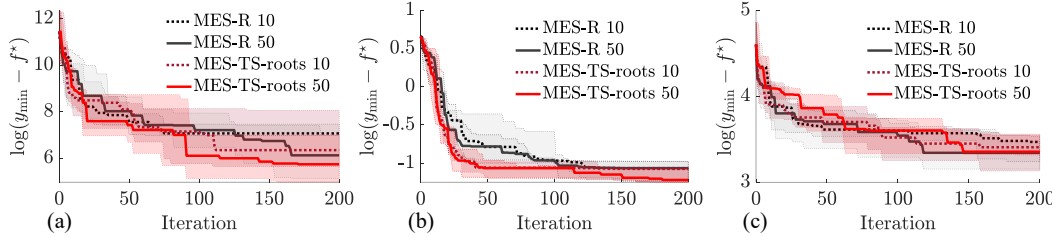

Figure 5: Performance of MES-R 10 and MES-R 50 for (a) the 4D Rosenbrock function, (b) the 6D Hartmann function, and (c) the 10D Levy function when TS-RF and TS-roots are used for generating random samples from $f^\star|\mathcal{D}$. The plots are histories of medians and interquartile ranges of solutions from ten runs of each method.

**TS-roots to Information-Theoretic Acquisition Functions.** We show how TS-roots can enhance the performance of MES (Wang & Jegelka, 2017), which uses information about the maximum function value $f^\star$ for conducting BO. One approach to computing MES generates a set of GP posterior samples using TS-RF and subsequently optimizes the generated functions for samples of $f^\star$ using a gradient-based multistart optimizer with a large number of random starting points (Wang & Jegelka, 2017). We hypothesize that high-quality $f^\star$ samples can improve the performance of MES. Thus, we assign both TS-roots and TS-RF as the inner workings of MES and then compare the resulting optimal solutions. Note that the inner-loop optimization of MES, which strongly influences the optimization results, is not addressed by TS-roots.

Specifically, we minimize the 4D Rosenbrock, 6D Hartmann, and 10D Levy functions using four versions of MES, namely MES-R 10, MES-R 50, MES-TS-roots 10, and MES-TS-roots 50. Here, MES-R (Wang & Jegelka, 2017) and MES-TS-roots correspond to TS-RF and TS-roots, respectively, while 10 and 50 represent the number of random samples $f^\star$ generated for computing the MES acquisition function in each iteration.

Figure 5 shows the optimization histories for ten independent runs of each MES method. On the 4D Rosenbrock and 6D Hartmann functions, MES with TS-roots demonstrates superior optimization performance and faster convergence compared to MES with TS-RF, especially when 50 samples of $f^\star$ are generated. For the 10D Levy function, TS-roots outperforms TS-RF when using 10 samples of $f^\star$, while their performance is comparable when 50 samples are used.

**Performance of Sample-Average Posterior Functions.** We investigate how $\alpha_{\text{aTS}}(\mathbf{x})$ influences the outer-loop optimization results. For this, we set $N_c \in \{1, 10, 50, 100\}$ for TS-roots to optimize the 2D Schwefel, 4D Rosenbrock, and 6D Ackley functions. We observe that increasing $N_c$ from 1 to 10 improves TS-roots performance on the 2D Schwefel, 4D Rosenbrock, and 6D Ackley functions (see Figure 14 in Appendix I). However, further increases in $N_c$ from 10 to 50 and 100 result in slight declines in solution quality as TS-roots transitions to exploitation. These observations indicate that there is an optimal value of $N_c$ for each problem at which TS-roots achieves its best performance by balancing exploitation and exploration. However, identifying the optimal value to maximize the performance of $\alpha_{\text{aTS}}(\mathbf{x})$ for a particular optimization problem remains an open issue.

## 7 CONCLUSION AND FUTURE WORK

We presented TS-roots, a global optimization strategy for posterior sample paths. It features an adaptive selection of starting points for gradient-based multistart optimizers, combining exploration and exploitation. This strategy breaks the curse of dimensionality by exploiting the separability of Gaussian process priors. Compared with random multistart and a genetic algorithm, TS-roots consistently yields higher-quality solutions in optimizing posterior sample paths, across a range of input dimensions. It also improves the outer-loop optimization performance of GP-TS and information-theoretic acquisition functions such as MES for Bayesian optimization. For future work, we aim to extend TS-roots to other spectral representations per Bochner's theorem (Mutny & Krause, 2018; Hensman et al., 2018; Solin & Särkkä, 2020). We also plan to study the ways and the probability of TS-roots failing to find the global optimum, as well as the impact of subset sizes.

ACKNOWLEDGMENTS

We would like to thank Tian PAN for his help with the `maxk_sum` problem, and the organizers and the participants of the NeurIPS 2024 Workshop on Bayesian Decision-making and Uncertainty for their invaluable discussions and feedback on an earlier version of the paper. The authors are supported by the University of Houston through the SEED program no. 000189862.

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

# A CHARACTERIZING THE LOCAL MINIMA OF A SEPARABLE FUNCTION

## A.1 PROOF OF PROPOSITION 1: A REPRESENTATION OF THE SET OF LOCAL MINIMA

Proposition 1 broadly applies to separable functions on a hypercube. Consider a separable function $f(\mathbf{x}) = \prod_{i=1}^{d} f_i(x_i)$ with domain $\mathcal{X} = \prod_{i=1}^{d} [\underline{x}_i, \overline{x}_i]$, where $f_i \in C^1([\underline{x}_i, \overline{x}_i]; \mathbb{R})$. To simplify the discussion, we further assume that $f_i$ is twice differentiable at its interior critical points $\mathring{\Xi}_i$. The gradient of $f$ can be written as:

$$\nabla f(\mathbf{x}) = \left( f_i'(x_i) \cdot \prod_{j \neq i}^{d} f_j(x_j) \right)_{i=1}^{d} = \left( f(\mathbf{x}) \cdot \frac{f_i'(x_i)}{f_i(x_i)} \right)_{i=1}^{d} = f(\mathbf{x}) \cdot \mathbf{v}(\mathbf{x}), \tag{6}$$

where $\mathbf{v}(\mathbf{x}) = \left( f_i'/f_i \right)_{i=1}^{d} = \left( \frac{\mathrm{d}}{\mathrm{d}x_i} \log f_i \right)_{i=1}^{d}$. The Hessian of $f$ can be written as:

$$\nabla^2 f(\mathbf{x}) = \mathrm{diag}\left\{ f_i''(x_i) \prod_{j \neq i}^{d} f_j(x_j) \right\}_{i=1}^{d} + \left[ f_i'(x_i) f_j'(x_j) \prod_{k \neq i,j}^{j \neq i} f_k(x_k) \right]_{i \in d}^{j \neq i} = f(\mathbf{x}) \, \mathrm{diag}(\mathbf{s} + \mathbf{v}\mathbf{v}^{\mathsf{T}}), \tag{7}$$

where $\mathbf{s}(\mathbf{x}) = \left( f_i''/f_i - (f_i'/f_i)^2 \right)_{i=1}^{d} = \left( \frac{\mathrm{d}^2}{\mathrm{d}x_i^2} \log f_i \right)_{i=1}^{d}$.

An interior point $\mathbf{x} \in \mathrm{int}\, \mathcal{X} := \prod_{i=1}^{d} (\underline{x}_i, \overline{x}_i)$ is a strong local minimum of $f$ if and only if $\nabla f(\mathbf{x}) = 0$ and $\nabla^2 f(\mathbf{x}) > 0$. From eq. (6), the first condition is satisfied in any of the following three cases: (1) $f_i(x_i) \neq 0$ and $f_i'(x_i) = 0$ for all $i \in \{1, \cdots, d\}$; (2) $f_i(x_i) = 0$ for exactly one $i \in \{1, \cdots, d\}$ and $f_i'(x_i) = 0$; or (3) $f_i(x_i) = 0$ for all $i \in I \subseteq \{1, \cdots, d\}$ where $|I| \geq 2$.

In case (1), the Hessian eq. (7) reduces to $\nabla^2 f(\mathbf{x}) = f(\mathbf{x}) \cdot \mathrm{diag}\{f_i''(x_i)/f_i(x_i)\}_{i=1}^{d}$, which is positive definite if and only if one of the following holds: (i) $f(\mathbf{x}) > 0$ and $f_i(x_i)f_i''(x_i) > 0$, for all $i \in \{1, \cdots, d\}$; or (ii) $f(\mathbf{x}) < 0$ and $f_i(x_i)f_i''(x_i) < 0$, for all $i \in \{1, \cdots, d\}$.

In case (2), the Hessian reduces to an all-zero matrix except for the $i$th diagonal entry: $[\nabla^2 f(\mathbf{x})]_{i,i} = f_i''(x_i) \prod_{j \neq i} f_j(x_j)$. Even if this entry is positive, the Hessian is still positive semi-definite, which means that there is a continuum of weak local minima: $\{x_i\} \times \prod_{j \neq i} [\underline{x}_j, \overline{x}_j]$. Besides, this case requires $f_i$ and $f_i'$ to have an identical root, which an event with probability zero.

In case (3), let $g_i(r_i) := f_i(x_i + r_i)$ be a shifted version of $f_i$, $i \in \{1, \cdots, d\}$. Taylor expansion at $\mathbf{r} = 0$ gives $g_i(r_i) = 0 + g_i'(0) r_i + o(r_i)$ for all $i \in I$ and $g_j(r_j) = g_j(0) + O(r_j)$ for all $j \notin I$. We have $g(\mathbf{r}) := \prod_{i=1}^{d} g_i(r_i) = c \prod_{i \in I} r_i + o(\prod_{i \in I} r_i) \cdot O(\prod_{j \notin I} r_j)$, where $c = \prod_{i \in I} g_i'(0) \cdot \prod_{j \notin I} g_j(0) \neq 0$. This means that there is a continuum of saddle points: $\{x_i\}_{i \in I} \times \prod_{j \notin I} [\underline{x}_j, \overline{x}_j]$.

For a boundary point $\mathbf{x} \in \partial \mathcal{X} := \mathcal{X} \setminus \mathrm{int}\, \mathcal{X}$, we partition the index set $\{1, \cdots, d\}$ into $L, R$, and $I$ such that $x_i = \underline{x}_i$ for all $i \in L$, $x_i = \overline{x}_i$ for all $i \in R$, and $x_i \in (\underline{x}_i, \overline{x}_i)$ for all $i \in I$. Define $\nabla_J := (\partial_j)_{j \in J}$ for any subset $J$ of the indices. Then $\mathbf{x}$ is a strong local minimum of $f$ if and only if the following conditions hold: (a) $\mathbf{x}$ is a strong local minimum in $\{x_j\}_{j \notin I} \times \prod_{j \in I} [\underline{x}_j, \overline{x}_j]$; (b) $\nabla_L f(\mathbf{x}) > 0$; and (c) $\nabla_R f(\mathbf{x}) < 0$.

Condition (a) holds if any only if $\nabla_I f(\mathbf{x}) = 0$ and $\nabla_I^2 f(\mathbf{x}) > 0$. Based on the previous discussion on interior local minima, it is equivalent to: (i) $f(\mathbf{x}) > 0$ and $f_i(x_i)f_i''(x_i) > 0$, for all $i \in I$; or (ii) $f(\mathbf{x}) < 0$ and $f_i(x_i)f_i''(x_i) < 0$, for all $i \in I$.

From eq. (6), condition (b) is equivalent to: (i) $f(\mathbf{x}) > 0$ and $f_i(x_i)f_i'(x_i) > 0$, for all $i \in L$; or (ii) $f(\mathbf{x}) < 0$ and $f_i(x_i)f_i'(x_i) < 0$, for all $i \in L$.

Similarly, condition (c) is equivalent to: (i) $f(\mathbf{x}) > 0$ and $-f_i(x_i)f_i'(x_i) > 0$, for all $i \in R$; or (ii) $f(\mathbf{x}) < 0$ and $-f_i(x_i)f_i'(x_i) < 0$, for all $i \in R$.

Summarizing the above discussions, we see that there is a unified way to identify the set $\check{X}$ of all strong local minima of $f$, which is stated in Proposition 1. The discussion for the set $\widehat{X}$ of local maxima is the exactly the same, except that the signs are flipped. This also means that $\widehat{X}$ and $\check{X}$ form a partition of the union $\Xi^{(0)} \sqcup \Xi^{(1)}$ of the two tensor grids.

If $f_i$ is not twice differentiable at some interior critical point $x_i$, we may replace $f_i''(x_i) > 0$ with the statement that $x_i$ is a strong local minimum of $f_i$, and replace $f_i''(x_i) < 0$ with the statement

that $x_i$ is a strong local maximum of $f_i$. The rest of the discussion still follows. In practice, the differentiability of the prior sample is not an issue, because it is almost always approximated by a finite sum of analytic functions, which is again analytic.

## A.2 NUMBER OF LOCAL MINIMA OF A SEPARABLE FUNCTION

In proposition 1, each set of candidate coordinates $\Xi_i$ is partitioned into mixed type and mono type:

$$\Xi_i^{(1)} = \{\xi_{i,j} \in \Xi_i : f_i(\xi_{i,j}) h_i(\xi_{i,j}) < 0\}, \quad \Xi_i^{(0)} = \{\xi_{i,j} \in \Xi_i : f_i(\xi_{i,j}) h_i(\xi_{i,j}) > 0\}.$$

Another partition of $\Xi_i$ is by the sign of the corresponding component function value:

$$\Xi_i^- = \{\xi_{i,j} \in \Xi_i : f_i(\xi_{i,j}) < 0\}, \quad \Xi_i^+ = \{\xi_{i,j} \in \Xi_i : f_i(\xi_{i,j}) > 0\}.$$

These two partitions create a finer partition of $\Xi_i$ into four subsets:

$$\Xi_i^{-(1)} = \Xi_i^- \cap \Xi_i^{(1)}, \quad \Xi_i^{-(0)} = \Xi_i^- \cap \Xi_i^{(0)}, \quad \Xi_i^{+(1)} = \Xi_i^+ \cap \Xi_i^{(1)}, \quad \Xi_i^{+(0)} = \Xi_i^+ \cap \Xi_i^{(0)}.$$

Denote the sizes of mixed and mono type candidate coordinates as $n_i^{(1)} = |\Xi_i^{(1)}|$ and $n_i^{(0)} = |\Xi_i^{(0)}|$, then the sizes of the two tensor grids $\Xi^{(1)}$ and $\Xi^{(0)}$ can be written as:

$$N^{(1)} := |\Xi^{(1)}| = \prod_{i=1}^{d} n_i^{(1)}, \quad N^{(0)} := |\Xi^{(0)}| = \prod_{i=1}^{d} n_i^{(0)}.$$

Define signed sums as the sums of signs of function values on the two tensor grids:

$$S^{(1)} := \sum_{\boldsymbol{\xi} \in \Xi^{(1)}} \operatorname{sign}(f(\boldsymbol{\xi})), \quad S^{(0)} := \sum_{\boldsymbol{\xi} \in \Xi^{(0)}} \operatorname{sign}(f(\boldsymbol{\xi})).$$

We now derive efficient formulas to calculate these signed sums, using $S^{(1)}$ as an example. Denote each coordinate in $\Xi_i^{(1)}$ as $\xi_{i,j}^{(1)}$. Denote each point in $\Xi^{(1)}$ as $\boldsymbol{\xi}_J^{(1)} = (\xi_{i,J_i}^{(1)})_{i=1}^d$, where multi-index $J = (J_i)_{i=1}^d \in \Pi^{(1)} := \prod_{i=1}^d \{1, \cdots, n_i^{(1)}\}$. The signed sum $S^{(1)}$ can be written as:

$$S^{(1)} = \sum_{J \in \Pi^{(1)}} \operatorname{sign}(f(\boldsymbol{\xi}_J^{(1)})) = \sum_{J \in \Pi^{(1)}} \operatorname{sign}\left(\prod_{i=1}^{d} f_i(\xi_{i,J_i}^{(1)})\right)$$

$$= \sum_{J \in \Pi^{(1)}} \prod_{i=1}^{d} \operatorname{sign}(f_i(\xi_{i,J_i}^{(1)})) = \prod_{i=1}^{d} \sum_{j=1}^{n_i^{(1)}} \operatorname{sign}(f_i(\xi_{i,j}^{(1)}))$$

$$= \prod_{i=1}^{d} \left[ \sum_{j=1}^{n_i^{(1)}} 1(f_i(\xi_{i,j}^{(1)}) > 0) - \sum_{j=1}^{n_i^{(1)}} 1(f_i(\xi_{i,j}^{(1)}) < 0) \right] = \prod_{i=1}^{d} \left[ |\Xi_i^{+(1)}| - |\Xi_i^{-(1)}| \right].$$

A formula for $S^{(0)}$ can be derived analogously. Denote set sizes:

$$n_i^{-(1)} = |\Xi_i^{-(1)}|, \quad n_i^{-(0)} = |\Xi_i^{-(0)}|, \quad n_i^{+(1)} = |\Xi_i^{+(1)}|, \quad n_i^{+(0)} = |\Xi_i^{+(0)}|,$$

then the signed sums can be calculated as:

$$S^{(1)} = \prod_{i=1}^{d} (n_i^{+(1)} - n_i^{-(1)}), \quad S^{(0)} = \prod_{i=1}^{d} (n_i^{+(0)} - n_i^{-(0)}).$$

The sizes of negative and positive strong local minima of a separable function can be written as:

$$\check{N}^- := |\check{X}^-| = \sum_{\boldsymbol{\xi} \in \Xi^{(1)}} 1(f(\boldsymbol{\xi}) < 0) = \frac{1}{2}(N^{(1)} - S^{(1)}), \tag{8}$$

$$\check{N}^+ := |\check{X}^+| = \sum_{\boldsymbol{\xi} \in \Xi^{(0)}} 1(f(\boldsymbol{\xi}) > 0) = \frac{1}{2}(N^{(0)} + S^{(0)}).$$

Therefore, the size of the strong local minima of a separable function can be written as:

$$\check{N} := |\check{X}| = |\check{X}^-| + |\check{X}^+| = \frac{1}{2}(N^{(1)} + N^{(0)} - S^{(1)} + S^{(0)}). \tag{9}$$

## B  Ordering the Local Minima of a Separable Function

### B.1  Filtering a Tensor Grid for High Absolute Values of a Separable Function

The step one in Section 3.4 is equivalent to the following problem: given coordinates $Z_i = \{\zeta_{i,1}, \cdots, \zeta_{i,t_i}\}$ and components values $F_i = \{f_{i,1}, \cdots, f_{i,t_i}\}$, $i \in \{1, \cdots, d\}$, of a separable function $f(\mathbf{x}) = \prod_{i=1}^d f_i(x_i)$, find points $\boldsymbol{\zeta}$ such that $|f(\boldsymbol{\zeta})|$ are the $k$ largest in the tensor grid $Z = \prod_{i=1}^d Z_i$.

Because $\log|f(\mathbf{x})| = \log|\prod_{i=1}^d f_i(x_i)| = \sum_{i=1}^d \log|f_i(x_i)|$, we can solve this problem as follows: define two-dimensional arrays $F = [F_1, \cdots, F_d]$ and $A = \log|F|$, solve $S = \mathtt{maxk\_sum}(A, k)$, and return $\{\boldsymbol{\zeta} = (\zeta_{1,I_1}, \cdots, \zeta_{d,I_d}) : I \in S\}$. Here the $\mathtt{maxk\_sum}$ algorithms finds the combinations from $A$ that gives the $k$ largest sums, which is described next.

### B.2  Top Combinations with the Largest Sums

Consider this problem: given a two-dimensional array $A = [\mathbf{a}_1, \cdots, \mathbf{a}_d]$, $\mathbf{a}_i = [a_{i,1}, \cdots, a_{i,t_i}]$, with $a_{i,1} \geq \cdots \geq a_{i,t_i}$, $i \in \{1, \cdots, d\}$, find $k$ multi-indices of the form $I = [I_1, \cdots, I_d]$ such that the sums $s_I := \sum_{i=1}^d a_{i,I_i}$ are the $k$ largest among all combinations $I \in \prod_{i=1}^d \{1, \cdots, t_i\}$.

An exhaustive search is intractable because the number of all possible combinations grows exponentially as $\prod_{i=1}^d t_i$. Instead, we use a min-heap to efficiently keep track of the top $k$ combinations. A min-heap is a complete binary tree, where each node is no greater than its children. The operations of inserting an element and removing the smallest element from a min-heap can be done in logarithmic time. Algorithm 2 gives a procedure to solve the above problem using min-heaps.

---

**Algorithm 2** $\mathtt{maxk\_sum}$: Combinations with the $k$ largest sums

---

**Input:** two-dimensional array $A$; number of top combinations $k$.
 1: Make the array nonpositive by replacing $\mathbf{a}_i$ with $\mathbf{a}_i - a_{i,1}\mathbf{1}$ for $i = 1, \cdots, d$.
 2: Create a min-heap by adding the elements of $\mathbf{a}_1$, each considered a combination of length one: index $I_1$, key $a_{1,I_1}$.
 3: At stage $i = 2, \cdots, d$: create a new min-heap consisting of length-$i$ combinations by adding each element in $\mathbf{a}_i$ to each combination in the min-heap at the previous stage: index $[I_1, \cdots, I_i]$, key $\sum_{j=1}^i a_{j,I_j}$. The size of the min-heap at each stage is capped at $k$ by popping the smallest sum from the min-heap when necessary.
**Output:** combinations in the min-heap at stage $d$.

---

This algorithm has time complexity $\mathcal{O}(tk \log k)$, where $t = \sum_{i=1}^d t_i \ll \prod_{i=1}^d t_i$, and space complexity $\mathcal{O}(dk)$. In TS-roots, the cost of $\mathtt{maxk\_sum}$ is small compared with the gradient-based multistart optimization of the posterior sample.

## C  Algorithms for TS-roots

### C.1  Spectral Sampling of Separable Gaussian Process Priors

Per Mercer's theorem on probability spaces (see e.g., Rasmussen & Williams (2006), Sec 4.3), any positive definite covariance function that is essentially bounded with respect to some probability measure $\mu$ on a compact domain $\mathcal{X}$ has a spectral representation $\kappa(\mathbf{x}, \mathbf{x}') = \sum_{k=0}^\infty \lambda_k \phi_k(\mathbf{x}) \phi_k(\mathbf{x}')$, where $(\lambda_k, \phi_k(\mathbf{x}))$ is a pair of eigenvalue and eigenfunction of the kernel integral operator. The corresponding GP prior sample can be written as $f_\omega(\mathbf{x}) = \sum_{k=0}^\infty w_k \sqrt{\lambda_k} \phi_k(\mathbf{x})$, where $w_k \overset{\text{iid}}{\sim} \mathcal{N}(0,1)$ are independent standard Gaussian random variables. Similar spectral representations exist per Bochner's theorem, which may have efficient discretizations (Solin & Särkkä, 2020; Mutny & Krause, 2018).

Given spectral representations of the univariate covariance functions of a separable Gaussian Process prior, we can accurately approximate the prior sample as:

$$f_\omega(\mathbf{x}) \approx \prod_{i=1}^{d} f_i(x_i; \omega_i), \quad f_i(x_i; \omega_i) \approx \sum_{k=0}^{N_i-1} w_{i,k} \sqrt{\lambda_{i,k}} \phi_{i,k}(x_i). \tag{10}$$

Here $N_i$ is selected for each variable such that $\lambda_{i,N_i-1}/\lambda_{i,1} \leq \eta_i$, where $\eta_i$ is sufficiently small (see Appendix H for the value used in this study). The product of the univariate sample functions forms a *third-order approximation* of the multivariate sample function; their moment functions differ at order four and higher even orders. Using spectral representations of the univariate components as in eq. (10) is much more efficient than directly using a spectral representation of the separable GP prior, because the former uses $\sum_{i=1}^{d} N_i$ univariate terms to approximate $\prod_{i=1}^{d} N_i$ multivariate terms in the latter. Another benefit of this representation is that the sample functions are separable, which can be exploited in downstream tasks.

**Spectrum of the Squared Exponential Covariance Function.** The univariate squared exponential (SE) covariance function can be written as $\kappa(x, x'; l) = \exp(-\frac{1}{2}s^2)$, where the relative distance $s = |x - x'|/l$ and length scale $l \in (0, \infty)$. The spectral representation of such a covariance function per Mercer's theorem is $\kappa(x, x') = \sum_{k=0}^{\infty} \lambda_k \phi_k(x) \phi_k(x')$. With a Gaussian measure $\mu = \mathcal{N}(0, \sigma^2)$ over the domain $\mathcal{X} = \mathbb{R}$, we can write the eigenvalues $\lambda_k$ and eigenfunctions $\phi_k(x)$ of the kernel integral operator as follows. (See e.g., Zhu et al. (1998) Sec. 4 and Gradshteyn & Ryzhik (2014) 7.374 eq. 8.)

Define constants $a = (2\sigma^2)^{-1}$, $b = (2l)^{-1}$, $c = \sqrt{a^2 + 4ab}$, and $A = \frac{1}{2}a + b + \frac{1}{2}c$. For $k \in \mathbb{N}$, the $k$th eigenvalue is $\lambda_k = \sqrt{\frac{a}{A}}\left(\frac{b}{A}\right)^k$ and the corresponding eigenfunction is $\phi_k(x) = \left(\frac{\pi c}{a}\right)^{1/4} \psi_k(\sqrt{c}x) \exp\left(\frac{1}{2}ax^2\right)$, where $\psi_k(x) = \left(\pi^{1/2}2^k k!\right)^{-1/2} H_k(x) \exp\left(-\frac{1}{2}x^2\right)$ and $H_k(x)$ the $k$th-order Hermite polynomial defined by $H_k(x) = (-1)^k \exp(x^2) \frac{d^k}{dx^k} \exp(-x^2)$.

Figure 6 shows approximations to the SE covariance function by truncated spectral representations with the first $N$ eigenpairs and by random Fourier features (Rahimi & Recht, 2007) with $N$ basis functions. The spectral representation per Mercer's theorem converges quickly to the true covariance function, while the random Fourier features representation requires a large number of basis functions and is inaccurate for $N < 1000$.

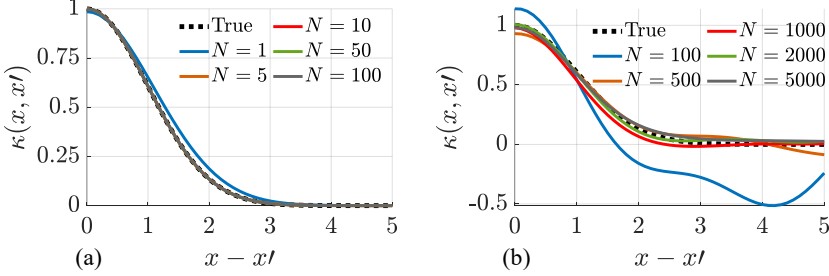

Figure 6: Approximate SE covariance functions from (a) the spectral representation per Mercer's theorem with the first $N$ eigenpairs and (b) the random Fourier features representation with $N$ basis functions. The plots are generated for $l = 1$.

## C.2 UNIVARIATE GLOBAL ROOTFINDING

Algorithm 3 outlines a method for univariate global rootfinding on an interval by solving an eigenvalue problem. When the orthogonal polynomial basis is the Chebyshev polynomials, the corresponding comrade matrix is called a colleague matrix, and we have the following theorem:

**Theorem 1** *Let $p(x) = \sum_{k=0}^{m} a_k T_k(x)$, $a_m \neq 0$, be a polynomial of degree $m$, where $T_k$ is the $k$th Chebyshev polynomial and $a_k$ is the corresponding weight. The roots of $p(x)$ are the eigenvalues of*

---

**Algorithm 3** `roots`: Univariate global rootfinding on an interval

---

**Input:** polynomial $p(x)$ of degree $m$ (or any real function $f(x)$)
 1: transform $p(x)$ into an orthogonal polynomial basis $p(x) = \sum_{k=0}^{m} a_k T_k(x)$
    (or approximate $f(x)$ on the interval using such a basis)
 2: solve all the eigenvalues of the comrade matrix $\mathbf{C}$ associated with the polynomial basis
**Output:** all the real eigenvalues $\{x_i\}_{i=1}^{r}$ in the interval, which are the roots of $p(x)$ (or $f(x)$)

---

*the following $m \times m$ colleague matrix:*

$$C = \begin{pmatrix} 0 & 1 & & & & \\ 1/2 & 0 & 1/2 & & & \\ & 1/2 & 0 & 1/2 & & \\ & & \ddots & \ddots & \ddots & \\ & & & & & 1/2 \\ & & & & 1/2 & 0 \end{pmatrix} - \frac{1}{2a_m} \begin{pmatrix} & & & & & \\ & & & & & \\ & & & & & \\ & & & & & \\ & & & & & \\ a_0 & a_1 & a_2 & \cdots & \cdots & a_{m-1} \end{pmatrix}, \quad (11)$$

*where the elements not displayed are zero.*

A proof of Theorem 1 is provided in Trefethen (2019), Chapter 18. A classical formula to compute the weights $\{a_k\}$ requires $\mathcal{O}(m^2)$ floating point operations, which can be reduced to $\mathcal{O}(m \log m)$ using a fast Fourier transform. Since the colleague matrix is tridiagonal except in the final row, the complexity of computing its eigenvalues can be improved from $\mathcal{O}(m^3)$ to $\mathcal{O}(m^2)$ operations, which can be further improved to $\mathcal{O}(m)$ via recursive subdivision of intervals (see Trefethen (2019)). Specifically, if $m > 100$, the interval is divided recursively so that on each subinterval the function can be accurately approximated by a polynomial of degree no greater than 100. The `roots` algorithm is implemented in the Chebfun package in MATLAB (Battles & Trefethen, 2004) and the chebpy package in Python (Richardson, 2016); both packages also implement other related programs such as `chebfun` for Chebyshev polynomial approximation and `diff` for differentiation.

### C.3 BEST LOCAL MINIMA OF A SEPARABLE FUNCTION

Given the univariate component functions of a separable function, Algorithm 4 finds the subset $S_{\mathrm{o}}$ of the local minima of the function with the $n_o$ smallest function values. This procedure requires the `maxk_sum` algorithm in Algorithm 2, the `roots` algorithm in Algorithm 3 and the related programs `chebfun` and `diff`, see also Appendix H.

In Algorithm 4, $\boldsymbol{\xi}, \mathbf{f}, \mathbf{h}, J, P$ are two-dimensional arrays, while $\mathbf{I}, \mathbf{\Pi}^{(1)}, \mathbf{\Pi}^{(0)}$ are matrices. Function evaluations at Lines 9, 10, 14, 19, 20, and 24 are only notational: the sign and value of the function can be computed efficiently by multiplying the signs and values of its components at the selected coordinates. For example, the statement $f(\boldsymbol{\xi}^{(1)}(\mathbf{I})) < 0$ at Line 9 can be evaluated as `rowXor`$(P^{(1)}(\mathbf{I}))$, where $P^{(1)}$ is a two-dimensional array with $P_i^{(1)} = P_i(\neg J_i)$, $P^{(1)}(\mathbf{I})$ is a matrix with $d$ columns, and `rowXor` is row-wise exclusive or operation. Similarly, the statement $f(\boldsymbol{\xi}^{(1)}(\mathbf{I}))$ at Line 10 can be evaluated as `rowProd`$(\mathbf{f}^{(1)}(\mathbf{I}))$, where `rowProd` is row products.

### C.4 DECOUPLED SAMPLING FROM GAUSSIAN PROCESS POSTERIORS

The decoupled sampling method for GP posteriors (Wilson et al., 2020), together with the spectral sampling of separable GP priors, is outlined in Algorithm 5.

### C.5 COMPUTATIONAL COMPLEXITY OF TS-ROOTS

Per Algorithm 1, the computational cost of the `TS-roots` method is dominated by a few tasks: (1) one call of `minsort` (Algorithm 4); (2) $n_{\mathrm{o}} + n$ evaluations of the posterior sample path $\widetilde{f}(\cdot)$; and (3) $n_{\mathrm{e}} + n_{\mathrm{x}}$ calls of the gradient-based optimizer `minimize`.

First, consider task (2). Evaluating $\widetilde{f}(\cdot)$ involves evaluating: (i) the prior sample path $f(\cdot)$, which involves evaluating its $d$ univariate component functions, each with a cost that depends on its spectral

---

**Algorithm 4** `minsort`: Best local minima of a separable function

---

**Input:** separable function $f(\mathbf{x}) = \prod_{i=1}^{d} f_i(x_i)$; set size $n_\mathrm{o}$; buffer coefficient $\alpha$ (defaults to 3).

1: $f_i(x_i) \leftarrow \texttt{chebfun}(f_i(x_i)), i = 1, \cdots, d$ ▷ Construct `chebfuns` for univariate components
   $f_i'(x_i) \leftarrow \texttt{diff}(f_i(x_i)); \quad f_i''(x_i) \leftarrow \texttt{diff}(f_i'(x_i))$ ▷ Compute first and second derivatives
2: $\{\xi_{i,j}\}_{j=1}^{r_i} \leftarrow \texttt{roots}(f_i'(x_i)), i = 1, \cdots, d$ ▷ Univariate global rootfinding
   $\{\xi_{i,0}\} \leftarrow \underline{x}_i; \quad \{\xi_{i,r_i+1}\} \leftarrow \overline{x}_i$ ▷ Include interval lower and upper bounds
   $\boldsymbol{\xi}_i \leftarrow [\xi_{i,0}, \xi_{i,1}, \cdots, \xi_{i,r_i}, \xi_{i,r_i+1}]^\mathsf{T}$ ▷ Candidate coordinate values $\{\Xi_i\}$
3: $\mathbf{f}_i \leftarrow f_i(\boldsymbol{\xi}_i), i = 1, \cdots, d$ ▷ Univariate function values
   $h_{i,j} \leftarrow f_i''(\xi_{i,j}), j = 1, \cdots, r_i$ ▷ Univariate second derivatives at critical points
   $h_{i,0} \leftarrow f_i'(\xi_{i,0}); \quad h_{i,r_i+1} \leftarrow -f_i'(\xi_{i,r_i+1})$ ▷ Univariate inward derivatives at interval ends
4: $J_i \leftarrow (\mathbf{f}_i \circ \mathbf{h}_i > 0); \quad P_i \leftarrow (\mathbf{f}_i > 0)$ ▷ Boolean vectors of sign parity and positivity
5: $\boldsymbol{\xi}_i^{(0)} \leftarrow \boldsymbol{\xi}_i(J_i); \quad \boldsymbol{\xi}_i^{(1)} \leftarrow \boldsymbol{\xi}_i(\neg J_i)$ ▷ Mono and mixed type candidate coordinates: $\Xi_i^{(0)}, \Xi_i^{(1)}$
   $\mathbf{f}_i^{(0)} \leftarrow \mathbf{f}_i(J_i); \quad \mathbf{f}_i^{(1)} \leftarrow \mathbf{f}_i(\neg J_i)$ ▷ Values at mono and mixed type candidate coordinates
6: $n_i^{(0)} \leftarrow \texttt{sum}(J_i); \quad n_i^{(1)} \leftarrow \texttt{sum}(\neg J_i)$
   $n_i^{+(0)} \leftarrow \texttt{sum}(P_i \,\&\, J_i); \quad n_i^{-(0)} \leftarrow \texttt{sum}((\neg P_i) \,\&\, J_i)$
   $n_i^{+(1)} \leftarrow \texttt{sum}(P_i \,\&\, (\neg J_i)); \quad n_i^{-(1)} \leftarrow \texttt{sum}((\neg P_i) \,\&\, (\neg J_i))$
   $N^{(0)} \leftarrow \prod_{i=1}^{d} n_i^{(0)}; \quad N^{(1)} \leftarrow \prod_{i=1}^{d} n_i^{(1)}$ ▷ Sizes of tensor grids
   $S^{(0)} \leftarrow \prod_{i=1}^{d}(n_i^{+(0)} - n_i^{-(0)}); \quad S^{(1)} \leftarrow \prod_{i=1}^{d}(n_i^{+(1)} - n_i^{-(1)})$ ▷ Signed sums
7: **if** $n_\mathrm{o} \le \breve{N}^- = \frac{1}{2}(N^{(1)} - S^{(1)})$ **then**
8: $\quad [\mathbf{s}, \mathbf{I}] \leftarrow \texttt{maxk\_sum}\left(\{\log(|\mathbf{f}_i^{(1)}|)\}_{i=1}^d, \alpha n_\mathrm{o}\right)$ ▷ The $\alpha n_\mathrm{o}$ largest $|f|$ in $\Xi^{(1)}$
9: $\quad \mathbf{I} \leftarrow \mathbf{I}[f(\boldsymbol{\xi}^{(1)}(\mathbf{I})) < 0, :]$ ▷ Multi-indices of best negative local minima
10: $\quad [\mathbf{b}, I] \leftarrow \texttt{mink}(f(\boldsymbol{\xi}^{(1)}(\mathbf{I})), n_\mathrm{o})$ ▷ The $n_\mathrm{o}$ smallest $f$ in $\breve{X}^-$
11: $\quad S_\mathrm{o} \leftarrow S_\mathrm{o}^- = \boldsymbol{\xi}^{(1)}(\mathbf{I}[I, :])$
12: **else**
13: $\quad \boldsymbol{\Pi}^{(1)} \leftarrow \prod_{i=1}^{d}\{1, \cdots, n_i^{(1)}\}$ ▷ Matrix of index combinations
14: $\quad \breve{\mathbf{I}}^- \leftarrow \boldsymbol{\Pi}^{(1)}[f(\boldsymbol{\xi}^{(1)}(\boldsymbol{\Pi}^{(1)})) < 0, :]$ ▷ Multi-indices of negative local minima
15: $\quad [\mathbf{b}, I] \leftarrow \texttt{sort}(f(\boldsymbol{\xi}^{(1)}(\breve{\mathbf{I}}^-)))$ ▷ Sort values in ascending order
16: $\quad \breve{X}^- \leftarrow \boldsymbol{\xi}^{(1)}(\breve{\mathbf{I}}^-[I, :])$ ▷ Negative local minima
17: $\quad$ **if** $n_\mathrm{o} \le \breve{N} = \frac{1}{2}(N^{(1)} - S^{(1)} + N^{(0)} + S^{(0)})$ **then**
18: $\quad\quad [\mathbf{s}, \mathbf{I}] \leftarrow \texttt{maxk\_sum}\left(\{\log(|\mathbf{f}_i^{(0)}|)\}_{i=1}^d, \alpha(n_\mathrm{o} - \breve{N}^-)\right)$ ▷ Largest $|f|$ in $\Xi^{(0)}$
19: $\quad\quad \mathbf{I} \leftarrow \mathbf{I}[f(\boldsymbol{\xi}^{(0)}(\mathbf{I})) > 0, :]$ ▷ Multi-indices of best positive local minima
20: $\quad\quad [\mathbf{b}, I] \leftarrow \texttt{mink}(f(\boldsymbol{\xi}^{(0)}(\mathbf{I})), n_\mathrm{o} - \breve{N}^-)$ ▷ The $n_\mathrm{o} - \breve{N}^-$ smallest $f$ in $\breve{X}^+$
21: $\quad\quad S_\mathrm{o} \leftarrow \breve{X}^- \bigcup S_\mathrm{o}^+, \ S_\mathrm{o}^+ = \boldsymbol{\xi}^{(0)}(\mathbf{I}[I, :])$
22: $\quad$ **else**
23: $\quad\quad \boldsymbol{\Pi}^{(0)} \leftarrow \prod_{i=1}^{d}\{1, \cdots, n_i^{(0)}\}$ ▷ Matrix of index combinations
24: $\quad\quad \breve{\mathbf{I}}^+ \leftarrow \boldsymbol{\Pi}^{(0)}[f(\boldsymbol{\xi}^{(0)}(\boldsymbol{\Pi}^{(0)})) > 0, :]$ ▷ Multi-indices of positive local minima
25: $\quad\quad [\mathbf{b}, I] \leftarrow \texttt{sort}(f(\boldsymbol{\xi}^{(0)}(\breve{\mathbf{I}}^+)))$ ▷ Sort values in ascending order
26: $\quad\quad S_\mathrm{o} \leftarrow \breve{X}^- \bigcup \breve{X}^+, \ \breve{X}^+ = \boldsymbol{\xi}^{(0)}(\breve{\mathbf{I}}^+[I, :])$ ▷ All local minima
27: $\quad$ **end if**
28: **end if**
**Output:** $S_\mathrm{o}$ ▷ Candidate exploration set: smallest $n_\mathrm{o}$ local minima in ascending order

---

representation (Appendix C.1); and (ii) the canonical basis $\boldsymbol{\kappa}_{\cdot,n}(\cdot)$, which costs $\mathcal{O}(dn)$ flops. When the data size $n$ is large, we can pre-filter the observed locations $X$ by the observations $\mathbf{y}$, which is a good estimate of $\widetilde{f}(X)$ depending on the observation noise. We assume that the number of observed locations after filtering is at most comparable to $n_\mathrm{e}$. The cost of task (2) is thus $\mathcal{O}(n_\mathrm{o}dn)$ flops.

Now consider task (3). Evaluating the gradient of $\widetilde{f}(\cdot)$ involves evaluating the gradients of $f(\cdot)$ and $\boldsymbol{\kappa}_{\cdot,n}(\cdot)$. Since both $f(\cdot)$ and $\boldsymbol{\kappa}_{\cdot,n}(\cdot)$ are separable functions, their gradients can be computed at a

---

**Algorithm 5** Decoupled sampling of Gaussian process posterior

---

**Input:** eigenpairs $\{(\lambda_{i,k}, \phi_{i,k}(x))\}_{i=1,\cdots,d}^{k=0,\cdots,N_i-1}$, data $\mathcal{D} = \{(\mathbf{x}^j, y^j)\}_{j=1}^n$, covariance matrix $\mathbf{C} = \mathbf{K}_{n,n} + \mathbf{\Sigma}$, canonical basis $\boldsymbol{\kappa}_{\cdot,n}(\mathbf{x}) = (\kappa(\mathbf{x}, \mathbf{x}^j))_{j=1}^n$.

1: $w_{i,k} \overset{\text{iid}}{\sim} \mathcal{N}(0, 1)$ ▷ Random coefficients for the prior sample
2: $f_\omega(\mathbf{x}) = \prod_{i=1}^d \sum_{k=0}^{N_i-1} w_{i,k} \sqrt{\lambda_{i,k}} \phi_{i,k}(x_i)$ ▷ Approximate prior sample
3: $\mathbf{f}_n \leftarrow [f_\omega(\mathbf{x}^1), \cdots, f_\omega(\mathbf{x}^n)]^\intercal$ ▷ Values of prior sample at observed locations
4: $\boldsymbol{\varepsilon} \sim \mathcal{N}_n(0, \mathbf{\Sigma})$ ▷ Random noise for the posterior sample
5: $\mathbf{v} \leftarrow \mathbf{C}^{-1}(\mathbf{y} - \mathbf{f}_n - \boldsymbol{\varepsilon})$ ▷ Linear solve via factorization (e.g., Cholesky or SVD)

**Output:** $\widetilde{f}_{\widetilde{\omega}}(\mathbf{x}) = f_\omega(\mathbf{x}) + \mathbf{v}^\intercal \boldsymbol{\kappa}_{\cdot,n}(\mathbf{x})$ ▷ Approximate posterior sample

---

cost comparable to evaluating their function values. Let $N_{\text{grad}}$ be the number of gradient evaluations required by the gradient-based optimizer. The cost of task (3) is thus $\mathcal{O}((n_e + n_x)N_{\text{grad}}dn)$ flops.

For task (1), the cost of the `minsort` algorithm is dominated by: (i) $d$ calls to `chebfun`, which evaluates the univariate components $f_i(\cdot)$ at a number of points depending on their complexity; (ii) $d$ calls to `roots` (Algorithm 3), which scales linearly with the polynomial degree $m$ of the chebfun object, itself dependent on the complexity of $f_i(\cdot)$; and (iii) at most one call to `maxk_sum` (Algorithm 2) at a cost of $\mathcal{O}(tn_o \log n_o)$, where $t = \sum_{i=1}^d t_i$ and $t_i$ is two plus the number of critical points of $f_i(\cdot)$ which depends on the complexity of $f_i(\cdot)$. The complexity of $f_i(\cdot)$, for the SE kernel for example (see Appendix C.1), can be quantified as the inverse length scale $\theta_i = 1/l_i$. We may define an average complexity as $\theta = \frac{1}{d}\sum_{i=1}^d \theta_i$. The cost of task (1) is thus $\mathcal{O}(d\theta n_o \log n_o)$ flops.

As explained in Appendix D, we can set $n_e$ and $n_x$ to small values and $n_o$ to a moderate value, independent of $\widetilde{f}(\cdot)$ and thus independent of $d$, $n$ and $\theta$. The overall cost of `TS-roots` thus scales as $\mathcal{O}(dn + d\theta)$, which is linear in the input dimension $d$.

## D   MINIMUM SIZE OF EXPLORATION AND EXPLOITATION SETS

We conduct an empirical experiment to determine minimal values for $n_e$ and $n_x$ of TS-roots algorithm, which are the sizes of $S_e$ and $S_x$, respectively. From this experiment, we also recommend a value for $n_o$, which is the size of $S_o$. Recall that points in $S_o$ are sorted in ascending order of prior sample values, while those in $S_e$ and $S_x$ are sorted in ascending order of posterior sample values.

Let $I_e$ and $I_x$ be the sets of indices of points in $S_e$ and $S_x$ that converge to the best local minimum of the posterior sample in each optimization iteration, respectively. Let $I_o$ be the set of indices of points in $S_o$ associated with $S_e$. Our hypothesis is that we have a high chance of finding a small index value in either $I_e$ or $I_x$. If this hypothesis is confirmed, then we can set both $n_e$ and $n_x$ at very small values, which significantly accelerate the inner-loop optimization. To confirm our hypothesis, we employ the following two steps. First, we set $n_e$ and $n_x$ at large values to mimic the effect of removing the set size limits, ensuring accurate solution of the global optimization problem. Then, we show that we have a high chance of finding a small index value from $I_e$ and/or $I_x$ in each optimization iteration.

We test our hypothesis on the 2D Schwefel, 4D Rosenbrock, 10D Levy, 16D Ackley, 16D Powell functions. We set $n_o = 5000$, $n_e = n_x = 1000$, and $\alpha = 3$ (buffer coefficient). The left and middle columns of Figure 7 show the smallest index values and the variation of index values from $I_e$ and/or $I_x$ of starting points that converge to the best local minimum $\mathbf{x}^\star$ of the posterior sample path in each optimization iteration. The left column also plots the index values from $I_o$ corresponding to the smallest index values from $I_e$, if exists. The right column shows the histograms of the smallest index values from $I_e$ and $I_x$ for all iterations considered. These results show that we have a high chance of finding a small index value from $I_e$ and/or $I_x$ in each iteration. This confirms our hypothesis. In fact, using the first point in $S_e$ and the first point in $S_e$—only two points—we can discover the global optimum most of the time. Interestingly, the smallest index values appear largely independent of both the optimization iteration and the input dimension. Furthermore, the results suggest that it is safe to set $n_o = 500$; and for almost exact global optimization, we suggest setting $n_e = 25$ and $n_x = 50$.

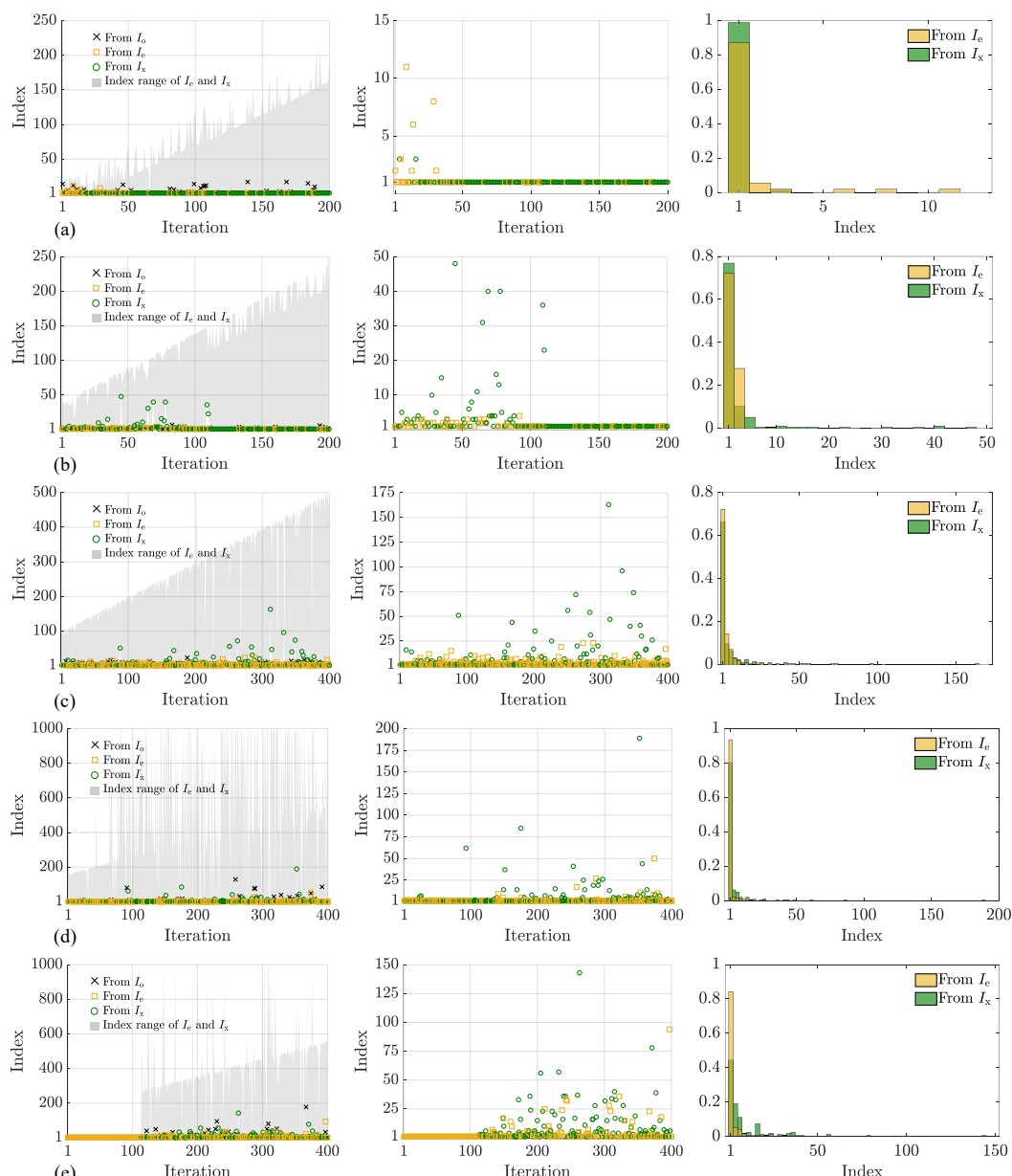

Figure 7: *Left column:* Minimum index values and index range of $I_e$ and/or $I_x$ for starting points that converge to the best local minimum $\mathbf{x}^\star$ of posterior sample in each optimization iteration, and index values of $I_o$ associated with minimum index values from $I_e$. *Middle column:* Zoom-in plots of index values. *Right column:* Historam of the minimum index values. (a) 2D Schwefel, (b) 4D Rosenbrock, (c) 10D Levy, (d) 16D Ackley, (e) 16D Powell functions.

## E   BAYESIAN OPTIMIZATION VIA THOMPSON SAMPLING

A general procedure for sequential optimization is given in Algorithm 6. The initial dataset $\mathcal{D}^0$ can either be empty or contain some observations. In the latter case we can write $\mathcal{D}^0 = \{(\mathbf{x}^i, y^i)\}_{i=1}^{n_0}$, where $n_0 \in \mathbb{N}_{>0}$. Three components of this algorithm can be customized: the observation model $\mathrm{Observe}(\mathbf{x})$, the optimization policy $\mathrm{Policy}(\mathcal{D})$, and the termination condition.

BO can be seen as an optimization policy for sequential optimization. A formal procedure is given in Algorithm 7. Three components of this algorithm can be customized: the prior probabilistic model

---

**Algorithm 6** Sequential optimization (Garnett, 2023)

---

**Input:** initial dataset $\mathcal{D}^0$
1: $k \leftarrow 1$
2: **repeat**
3:      $\mathbf{x}^k \leftarrow \text{Policy}(\mathcal{D}^{k-1})$
4:      $y^k \leftarrow \text{Observe}(\mathbf{x}^k)$
5:      $\mathcal{D}^k \leftarrow \mathcal{D}^{k-1} \cup \{(\mathbf{x}^k, y^k)\}$
6: **until** termination condition reached
**Output:** $\mathcal{D}$

---

**Algorithm 7** Bayesian optimization policy

---

**Input:** a prior stochastic process $f$ for the objective function $f_{\text{true}}$, current dataset $\mathcal{D}^{k-1}$
1: determine the posterior $f^k := f|\mathcal{D}^{k-1}$
2: derive an acquisition function $\alpha^k(\mathbf{x})$ from $f^k$
3: global optimization $\mathbf{x}^k \leftarrow \arg\min_{\mathbf{x}\in\mathcal{X}} \alpha^k(\mathbf{x})$
**Output:** $\mathbf{x}^k$

---

$f$, the acquisition function $\alpha$, and the global optimization algorithm. Any probabilistic model of the objective function $f_{\text{true}}$ can be seen as a probability distribution on a function space, and the prior $f$ is usually specified as a stochastic process such as a GP. The acquisition function $\alpha$ derived from the posterior $f|\mathcal{D}$ can be either deterministic—such as EI and LCB—or stochastic, such as GP-TS. To simplify notation, we state the global optimization problem of $\alpha(\mathbf{x})$ as minimization rather than maximization. The two problems are the same with a change of sign to the objective.

When applied to BO, GP-TS generates a random acquisition function simply by sampling the posterior model. That is, given the posterior $f^k$ at the $k$th BO iteration, the GP-TS acquisition function is a random function: $\alpha^k(\mathbf{x}) \sim f^k$.

## F    BENCHMARK FUNCTIONS

The analytical expressions for the benchmark functions used in Section 6 are given below. The global solutions of these functions are detailed in (Surjanovic & Bingham, 2013).

**Schwefel Function:**

$$f(\mathbf{x}) = 418.9829d - \sum_{i=1}^{d} x_i \sin\left(\sqrt{|x_i|}\right). \tag{12}$$

This function is evaluated on $\mathcal{X} = [-500, 500]^d$ and has a global minimum $f^\star := f(\mathbf{x}^\star) = 0$ at $\mathbf{x}^\star = [420.9687, \cdots, 420.9687]^\intercal$. This function is $C^1$ at $\mathbf{x} = 0$.

**Rosenbrock Function:**

$$f(\mathbf{x}) = \sum_{i=1}^{d-1} \left[100(x_{i+1} - x_i^2)^2 + (x_i - 1)^2\right]. \tag{13}$$

This function is evaluated on $\mathcal{X} = [-5, 10]^d$ and has a global minimum $f^\star = 0$ at $\mathbf{x}^\star = [1, \cdots, 1]^\intercal$.

**Levy Function:**

$$f(\mathbf{x}) = \sin^2(\pi w_1) + \sum_{i=1}^{d-1}(w_i - 1)^2\left[1 + 10\sin^2(\pi w_i + 1)\right] + (w_d - 1)^2\left[1 + \sin^2(2\pi w_d)\right], \tag{14}$$

where $w_i = 1 + \frac{x_i - 1}{4}$, $i = 1, \cdots, d$. This function is evaluated on $\mathcal{X} = [-10, 10]^d$ and has a global minimum $f^\star = 0$ at $\mathbf{x}^\star = [1, \cdots, 1]^\intercal$.

**Ackley Function:**

$$f(\mathbf{x}) = -a \exp\left(-b\sqrt{\frac{1}{d}\sum_{i=1}^{d} x_i^2}\right) - \exp\left(\frac{1}{d}\sum_{i=1}^{d}\cos(cx_i)\right) + a + \exp(1), \qquad (15)$$

where $a = 20$, $b = 0.2$, and $c = 2\pi$. This function is evaluated on $\mathcal{X} = [-10, 10]^d$ and has a global minimum $f^\star = 0$ at $\mathbf{x}^\star = [0, \cdots, 0]^\mathsf{T}$. This function not differentiable at $\mathbf{x}^\star$.

**Powell Function:**

$$f(\mathbf{x}) = \sum_{i=1}^{d/4}\left[(x_{4i-3} + 10x_{4i-2})^2 + 5(x_{4i-1} - x_{4i})^2 + (x_{4i-2} - 2x_{4i-1})^4 + 10(x_{4i-3} - x_{4i})^4\right].$$

$$(16)$$

This function is evaluated on $\mathcal{X} = [-4, 5]^d$ and has a global minimum $f^\star = 0$ at $\mathbf{x}^\star = [0, \cdots, 0]^\mathsf{T}$.

**6d Hartmann Function:**

$$f(\mathbf{x}) = -\sum_{i=1}^{4} a_i \exp\left(-\sum_{j=1}^{6} A_{ij}(x_j - P_{ij})^2\right), \qquad (17)$$

where

$$\mathbf{a} = [1, 1.2, 3, 3.2]^\mathsf{T}, \qquad (18a)$$

$$\mathbf{A} = \begin{bmatrix} 10 & 3 & 17 & 3.5 & 1.7 & 8 \\ 0.05 & 10 & 17 & 0.1 & 8 & 14 \\ 3 & 3.5 & 1.7 & 10 & 17 & 8 \\ 17 & 8 & 0.05 & 10 & 0.1 & 14 \end{bmatrix}, \qquad (18b)$$

$$\mathbf{P} = 10^{-4}\begin{bmatrix} 1312 & 1696 & 5569 & 124 & 8283 & 5886 \\ 2329 & 4135 & 8307 & 3736 & 1004 & 9991 \\ 2348 & 1451 & 3522 & 2883 & 3047 & 6650 \\ 4047 & 8828 & 8732 & 5743 & 1091 & 381 \end{bmatrix} \qquad (18c)$$

This function is evaluated on $\mathcal{X} = [0, 1]^6$ and has a global minimum $f^\star = -3.32237$ at $\mathbf{x}^\star = [0.20169, 0.150011, 0.476874, 0.275332, 0.311625, 0.6573]^\mathsf{T}$. The rescaled version $\tilde{f}(\mathbf{x}) = \frac{f(\mathbf{x})-2.58}{1.94}$ (Picheny et al., 2013) is used in the experiments.

## G  TEN-BAR TRUSS

Consider a ten-bar truss shown in Figure 8. The truss has ten members and is subjected to vertical load $P_1 = 60$ kN at node 2, vertical load $P_2 = 40$ kN at node 3, and horizontal load $P_3 = 10$ kN at node 3. The Young's modulus of the truss material $E = 200$ GPa. The length parameter $L = 1$ m. Let $A(\mathbf{x}) = \sum_{i=1}^{10} x_i$ and $\delta_3(\mathbf{x})$ denote the total area of the cross-sectional areas of the truss members and the vertical displacement at node 3, respectively, where $\mathbf{x} = [x_1, \ldots, x_{10}]^\mathsf{T}$ is the vector of cross-sectional areas of the truss members. The optimization problem formulated for the truss is to minimize both $A(\mathbf{x})$ and $\delta_3(\mathbf{x})$. Since $A(\mathbf{x})$ and $\delta_3(\mathbf{x})$ are competing, we define the objective function as a weight-sum of $A(\mathbf{x})$ and $\delta_3(\mathbf{x})$, such that

$$f(\mathbf{x}) = w_1 \frac{A(\mathbf{x})}{A_{\max}} + w_2 \frac{\delta_3(\mathbf{x})}{\delta_{\max}}, \qquad (19)$$

where $\mathbf{x} \in [1, 20]^{10}$ cm$^2$, $w_1 = 0.6$, $w_2 = 0.4$, $A_{\max} = 200$ cm$^2$, and $\delta_{\max} = 3$ cm.

## H  EXPERIMENTAL DETAILS

**Data Generation.** We generate 20 initial datasets for each problem. The input observations are randomly generated using the Latin hypercube sampling (Owen, 1992) within $[-1, 1]^d$, where $d$ represents the number of input variables. The normalized input observations are transformed into their real spaces to evaluate the corresponding objective function values which are then standardized using the $z$-score for processing optimization. Each BO method in comparison starts from each of the generated datasets.

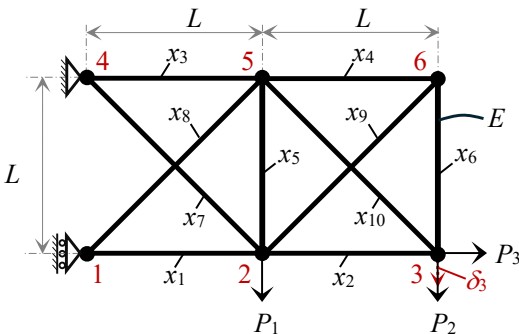

Figure 8: Ten-bar truss. Cross-sectional areas of ten truss members are the input variables $x_i$, $i \in \{1 \ldots, 10\}$. Known parameters include length $L$, Young's modulus of truss material $E$, and external loads $P_j$, $j \in \{1, 2, 3\}$. The vertical displacement at node 3 is denoted as $\delta_3$.

**Key Parameters for TS-roots and other BO Methods.** We use squared exponential (SE) covariance functions for our experiments. The spectra of univariate SE covariance functions for all problems (see Appendix C.1) are determined using the Gaussian measure $\mu = \mathcal{N}(0, 1)$. The number of terms $N_i$, $i \in \{1, \cdots, d\}$, of each truncated univariate spectrum is determined such that $\lambda_{i,N_i-1}/\lambda_{i,1} \leq \eta_i$, where $\eta_i = 10^{-16}$. If $N_i > 1000$, we set $N_i = 1000$ to trade off between the accuracy of truncated spectra and computational cost. We also set $n_o = 500$. The maximum size of the exploration set is $n_e = 250$. The maximum size of the exploitation set is $n_x = 200$.

The number of initial observations is $10d$ for all problems. The standard deviation of observation noise $\sigma_n = 10^{-6}$ is applied for standardized output observations. The number of BO iterations for the 2D Schwefel and 4D Rosenbrock functions is 200, while that for the 10D Levy, 16D Ackley, and 16D Powell functions is 800. Other GP-TS methods for optimization of benchmark test functions including TS-DSRF (i.e., TS using decoupled sampling with random Fourier features) and TS-RF (i.e., TS using random Fourier features) are characterized by a total of 2000 random Fourier features.

To ensure a fair comparison of outer-optimization results, we first implement TS-roots and record the number of starting points used in each optimization iteration. We then apply other BO methods, each employing a gradient-based multistart optimizer with the same number of random starting points and identical termination criteria as those used for TS-roots in each iteration.

For the comparative inner-loop optimization performance of the proposed method via rootfinding with the random multistart and genetic algorithm approaches, we set the same termination tolerance on the objective function value as the stopping criterion for the methods. In addition, the number of starting points for the random multistart and the population size of the genetic algorithm are the same as the number of points in both the exploration and exploitation sets of rootfinding in each optimization iteration.

**Computational Tools.** We carry out all experiments, except those for inner-loop optimization, using a designated cluster at our host institution. This cluster hosts 9984 Intel CPU cores and 327680 Nvidia GPU cores integrated within 188 compute and 20 GPU nodes. The inner-loop optimization is implemented on a PC with an Intel® Core™ i7-1165G7 @ 2.80 GHz and 16 GB memory.

For the univariate global rootfinding via Chebyshev polynomials, we use MATLAB's Chebfun package (Battles & Trefethen, 2004) and its corresponding implementation in Python, called chebpy (Richardson, 2016).

# I   ADDITIONAL RESULTS

**Comparison of TS-roots and LogEI Outer-loop Optimization Results.** Figure 9 compares the performance of the outer-loop optimization of TS-roots and LogEI (Ament et al., 2023) on the five benchmark functions and a real-world ten-bar truss structure. Across all examples, save 16D Powell, TS-roots outperforms LogEI, further demonstrating the robustness of our approach. The compar-

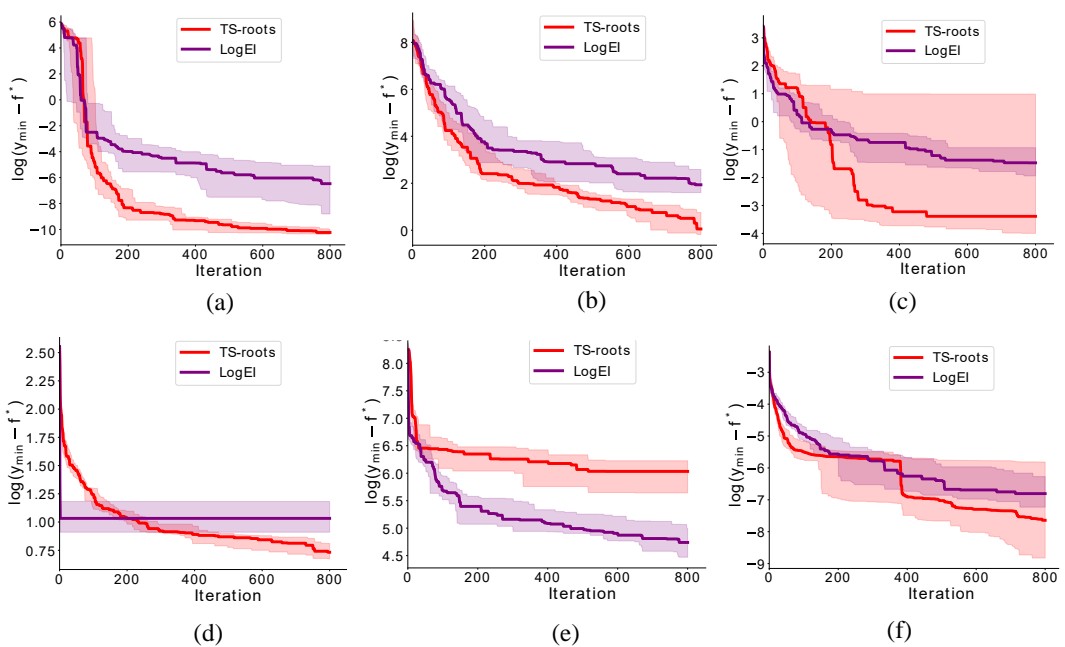

Figure 9: Outer-loop optimization results for the (a) 2D Schwefel function, (b) 4D Rosenbrock function, (c) 10D Levy function, (d) 16D Ackley function, (e) 16D Powell function, and (f) ten-bar truss problem. The plots are histories of medians and interquartile ranges of solution values from 20 runs of TS-roots and LogEI.

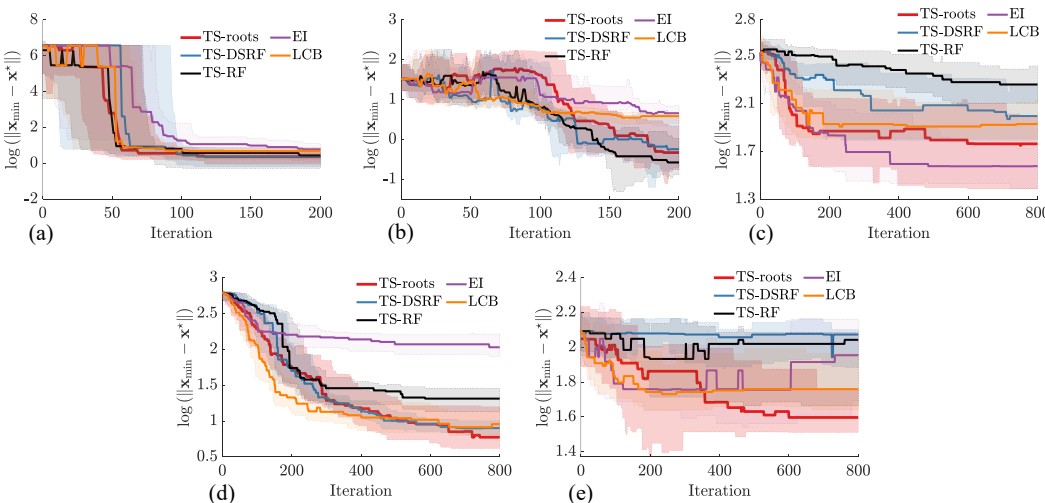

Figure 10: Outer-loop optimization results for (a) the 2D Schwefel, (b) 4D Rosenbrock, (c) 10D Levy, (d) 16D Ackley, (e) 16D Powell functions. The plots are histories of medians and interquartile ranges of solution locations from 20 runs of TS-roots, TS-DSRF (i.e., TS using decoupled sampling with random Fourier features), TS-RF (i.e., TS using random Fourier features), EI, and LCB.

atively weaker performance on the 16D Powell function can be attributed to its convex landscape, where BO generally underperforms relative to gradient-based methods.

**Distance to Global Minimum.** Figure 10 shows the solution locations from 20 runs of TS-roots, TS-DSRF, TS-RF, EI, and LCB for the 2D Schwefel, 4D Rosenbrock, 10D Levy, 16D Ackley, 16D Powell functions.

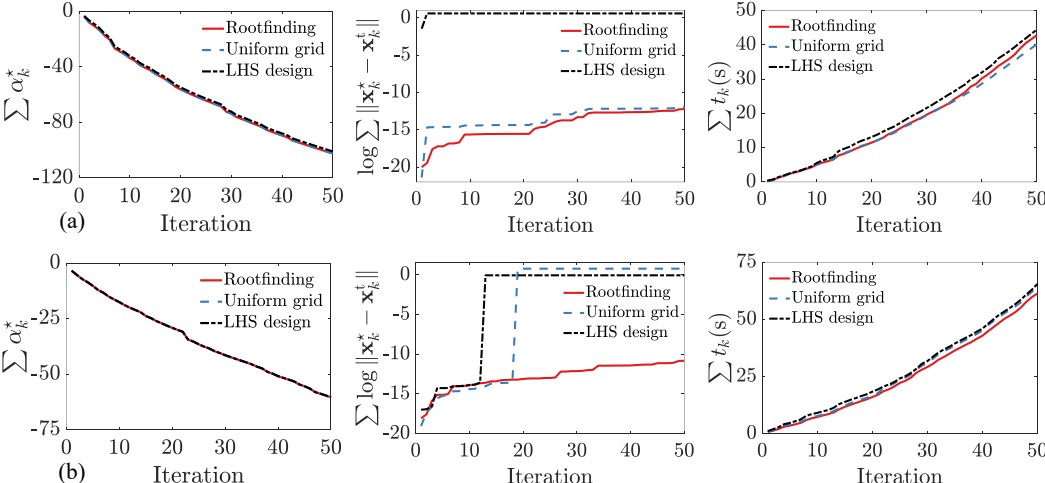

Figure 11: Inner-loop optimization results by three different initialization schemes, i.e., rootfinding, uniform grid, and Latin hypercube sampling, for (a) the 2D Schwefel and (b) 4D Rosenbrock functions. The plots are cumulative values of optimized GP-TS acquisition functions $\alpha_k^\star$, cumulative distances between new solution points $\mathbf{x}_k^\star$ and the true global minima $\mathbf{x}_k^t$ of the acquisition functions, and cumulative CPU times $t_k$ for optimizing the acquisition functions.

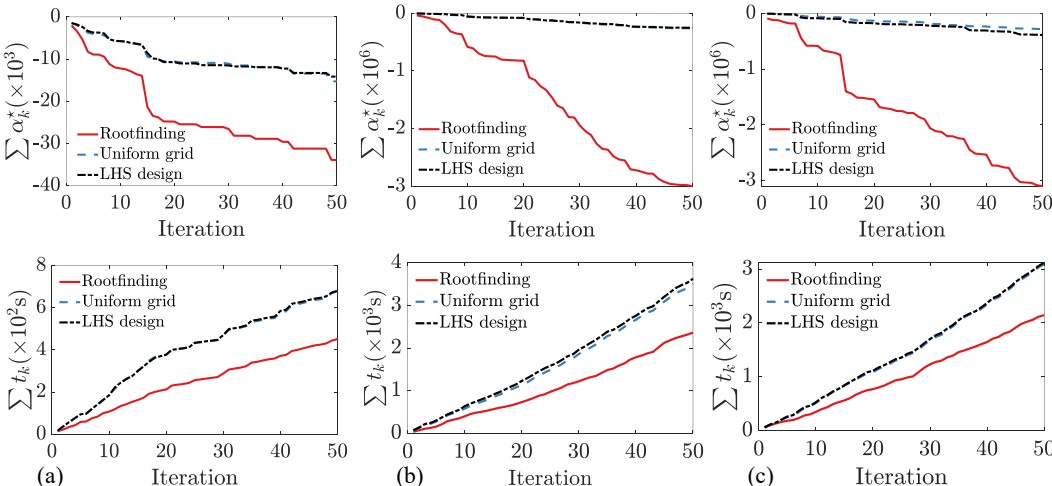

Figure 12: Inner-loop optimization results by three different initialization schemes, i.e., rootfinding, uniform grid, and Latin hypercube sampling, for (a) the 10D Levy, (b) 16D Ackley, and (c) 16D Powell functions. The plots are cumulative values of optimized GP-TS acquisition functions $\alpha_k^\star$ and cumulative CPU times $t_k$ for optimizing the acquisition functions.

**Comparison of Inner-loop Optimization Results.** Figures 11 and 12 compare the performance of the inner-loop optimization by three different initialization schemes, i.e., rootfinding, uniform grid, and Latin hypercube sampling, for low-dimensional cases of the 2D Schwefel and 4D Rosenbrock functions, and for higher-dimensional cases of the 10D Levy, 16D Ackley, and 16D Powell functions. Rootfinding performs better than the uniform grid and Latin hypercube sampling initialization schemes, especially in high-dimensional settings.

**Sample-average Posterior Function.** Figure 13 shows how we can improve the exploitation of GP-TS when increasing the exploration–exploitation control parameter $N_c$.

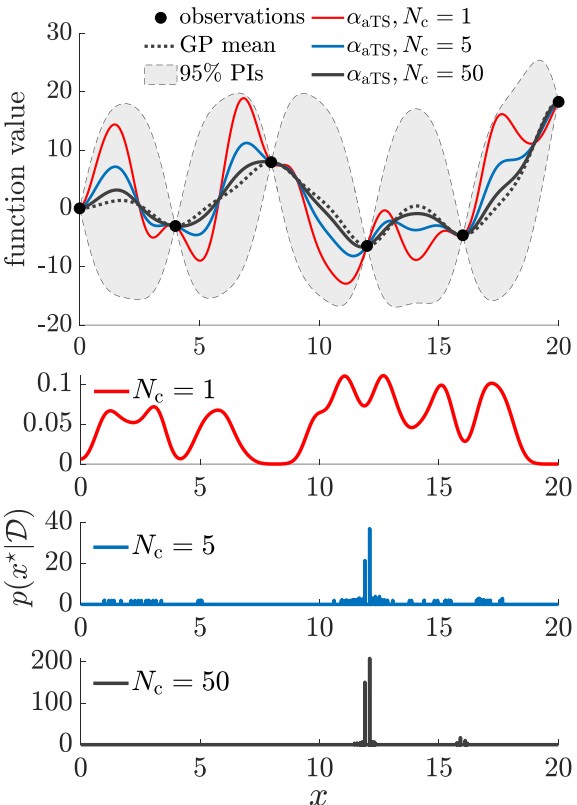

Figure 13: Sample-average posterior function for different values of $N_c$. The posterior function approaches the GP mean and the conditional distribution of the solution location $p(x^\star|\mathcal{D})$ is more concentrated when we increase $N_c$.

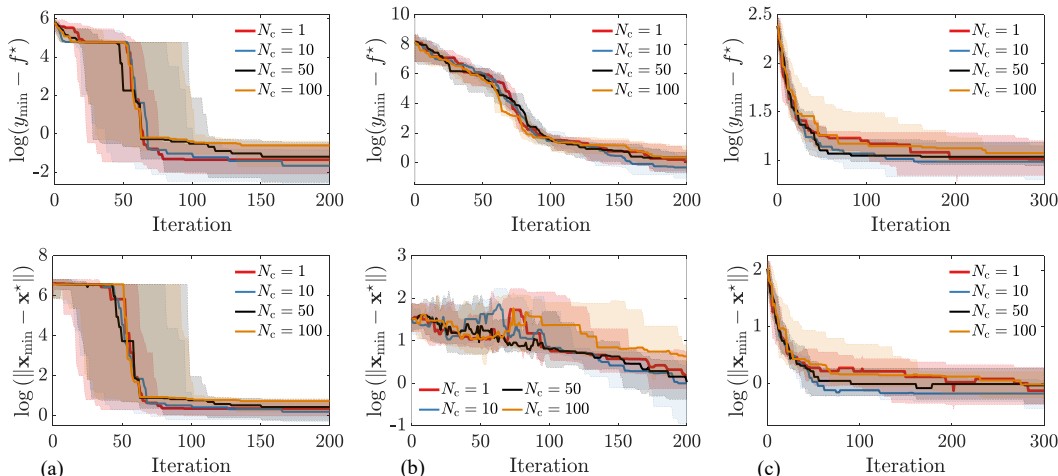

Figure 14: Performance of sample-average TS-roots with different control values $N_c$ for (a) the 2D Schwefel, 4D Rosenbrock, and (b) 6D Ackley functions. The plots are histories of medians and interquartile ranges of solution values and solution locations from 20 runs of TS-roots for each $N_c$ value.

**Performance of Sample-average TS-roots.** Figure 14 shows the performance of sample-average TS-roots with different exploration–exploitation control parameters $N_c$ for the 2D Schwefel, 4D Rosenbrock, and 6D Ackley functions.

