# OpenReview forum: "Optimizing Posterior Samples for Bayesian Optimization via Rootfinding"
_ICLR.cc/2025/Conference — ICLR 2025 Poster_

### Official Review · Reviewer_AkKy · 2024-10-19

**Soundness:** 3
**Presentation:** 3
**Contribution:** 3
**Rating:** 8
**Confidence:** 3

**Summary:**

The paper is about improving the inner-loop in Bayesian Optimization methods that rely on optimization of posterior samples. This is done by selecting better starting points for the gradient based optimization of the posterior samples. The set of starting points is made up from a subset of the local minima of the prior and the local minimal of the posterior mean function (the subset where the posterior values are highest, respectively). The empirical performance of the method is assessed on common benchmark functions.

**Strengths:**

* The method is motivated well (Section 3.2)
* The method performs better than (or at least on-par with ) the baselines empirically.
* I think it is good that also the outer-loop performance is tested.  At the first sight, I was wondering whether a potentially slight improvement in the inner loop would really matter for the final performance, but now I am more convinced that it does.
* Figure 1 is very useful

**Weaknesses:**

* For me, a bit more explanation on the decouple representation of GP Posteriors would have been useful. There is a reference to Wilson et al. 2020, but if one is not familiar with that particular piece of work, a bit more explanation on the decouple representation of GP Posteriors would have been useful. In particular, because this representation later on is key in the construction of the method.
* I think it would be nicer to either have Figure 8/Figure 9 from the Appendix near the "Sample-Average Posterior Function" Section. Or maybe even have the "Sample-Average Posterior Function" Section in the methods part instead of the results part? Otherwise, I reads more like an afterthought.

**Questions:**

* Is the method also applicable to Entropy Search [1]? The paragraph (l. 54-63) in the introduction mentions many information-theoretic acquisition functions besides this one and Entropy Search is arguably one of the main/first such acquisition functions.

[1] Entropy Search for Information-Efficient Global Optimization (Hennig and Schuler, 2012)

**Details Of Ethics Concerns:**

-

---

> ### Author Response · Authors · 2024-11-25
> **Response to Reviewer AkKy**
>
> Thank you for your constructive feedback on our paper!
> We have carefuly revised our paper, with some important new analytical and experimental results;
> please see the comment "Changes to the paper".
> Below are specific responses to the review.
>
> > For me, a bit more explanation on the decouple representation of GP Posteriors would have been useful. There is a reference to Wilson et al. 2020, but if one is not familiar with that particular piece of work, a bit more explanation on the decouple representation of GP Posteriors would have been useful. In particular, because this representation later on is key in the construction of the method.
>
> We agree that it would be helpful to further elaborate on the pathwise conditioning representation
> of Gaussian process posterior sample paths, even though it is not a contribution of this work.
> We have thus added the following additional details
> (see Section "2 General Background" paragraph "Decoupled Representation of GP Posteriors"):
>
> "This representation has its roots in Matheron’s update rule that transforms a joint distribution of Gaussian variables into a conditional one (see e.g., Hoffman & Ribak (1991)). This formula is exact, in that $\overset{\text{d}}{=}$ denotes equality in distribution, and it preserves the differentiability of the prior sample. It is also computationally efficient for posterior sampling: the cost is independent of the input dimension $d$, linear in the data size $n$ at evaluation time, and the weight vector for $\boldsymbol{\kappa}_{\cdot,n}(\mathbf{x})$ can be solved accurately using an iterative algorithm that scales linearly with $n$ (Lin et al., 2023)."
>
> References:
> - Hoffman Y and Ribak E. Constrained Realizations of Gaussian Fields: A Simple Algorithm.
>   Astrophysical Journal Letters, 380:L5–L8, October 1991.
> - Jihao Andreas Lin, Javier Antoran, Shreyas Padhy, David Janz, Jose Miguel Hernandez-Lobato, and
>   Alexander Terenin. Sampling from Gaussian process posteriors using stochastic gradient descent.
>   Advances in Neural Information Processing Systems, volume 36, pp. 36886–36912. 2023.
>
> > I think it would be nicer to either have Figure 8/Figure 9 from the Appendix near the "Sample-Average Posterior Function" Section. Or maybe even have the "Sample-Average Posterior Function" Section in the methods part instead of the results part? Otherwise, I reads more like an afterthought.
>
> Regarding the Sample-Average Posterior Function, we have created a new section for it (Section 4).
> While we intended to include related illustrations for clarity,
> the page limit for ICLR papers unfortunately prevents us from doing so.
>
> The paper is structured in the current format because its main contribution
> is the TS-roots algorithm. The sample-average posterior function is a nice observation
> based on the decouple representation of GP Posteriors, which is potentially useful for an explicit
> control of exploitation in Thompson sampling without sacrificing computational efficiency.
> The intention is to not distract readers from the main narrative of the paper.
>
> > Is the method also applicable to Entropy Search [1]? The paragraph (l. 54-63) in the introduction mentions many information-theoretic acquisition functions besides this one and Entropy Search is arguably one of the main/first such acquisition functions.
>
> Yes, we can use the TS-root algorithm to approximate
> the conditional distribution of the minimum location $p_{\min}$
> which is a key requirement for computing the Entropy Search acquisition function,
> see Line 4 of Algorithm 1 in [1].
> We also added a citation to Entropy Search (Hennig and Schuler, 2012) in the Introduction,
> where we cited a few other information-theoretic acquisition functions.
>
> References:
> - [1] Entropy Search for Information-Efficient Global Optimization (Hennig and Schuler, 2012).
>
> We hope our responses have addressed your concerns
> and that our revisions have significantly improved the paper.
> If so, we would greatly appreciate your consideration of increasing your scores.
> Please do not hesitate to let us know if you have any further questions or comments.

---

> > ### Comment · Reviewer_AkKy · 2024-11-25
> >
> > Thank you for the clarifications. They increase my confidence in this being a good paper. I will increase my confidence score from 2 to 3.

---

### Official Review · Reviewer_kcrQ · 2024-10-28

**Soundness:** 2
**Presentation:** 2
**Contribution:** 2
**Rating:** 6
**Confidence:** 4

**Summary:**

This paper focuses on global optimization of acquisition functions in the framework of Bayesian optimization (BO). The common practice of BO is to utilize routine algorithms, such as dividing rectangles (DIRECT),  covariance matrix adaptation evolution strategy (CMA-ES), and Thompson sampling (TS), to solve the acquisition function optimization problem. The authors here propose to leverage root finding procedures to improve/accelerate optimization of posterior sample-based acquisition functions. The basic idea is to select a subset of posterior samples as the starting points to conduct gradient-based root finding for searching local minima of the acquisition function. The global optimum is obtained by ranking the local optima. The performance of the proposed methods is evaluated by comparison with some standard BO algorithms on several benchmark testing functions.

**Strengths:**

1. This paper selects an uncommon topic of BO, namely global optimization of acquisition functions, and conduct a systematic research on it.
2. The paper proposes a novel strategy that uses root finding to search the global optimum of the acquisition functions, and evaluates it by numerical experiments.

**Weaknesses:**

1. The premise for that this paper is an important contribution to the community is that global optimization of acquisition functions is really necessary for guaranteeing BO's performance. Unfortunately this is not proved or even discussed in the paper. Is it worth to develop more complex algorithms which consume more computations to optimize the acquisition function? Is this better than we using the same resource to build a better surrogate model to approximate the objective function? Why the community paid more attention to the latter issue? It's likely because it plays a more important role for BO's performance.

2. As shown in Figure 2, for some cases, the proposed method, TS-roots, performs better than othe competitors; while for the other cases, it just performs on par with or even worse than the standard commonly used baseline BO methods. In addition, only several simple benchmark test objective functions are involved in the experiments. Combining the above two points, the experiment results don't provide enough evidence for that the proposed solution can bring remarkable benefits for solving challenging real-life problems.

**Questions:**

1. There are some hyperparameters, e.g., n_o, n_e, n_x, that control the balance between exploration and exploitation. Given a specific optimization problem to be resolved, how to specify appropriate values for these hyperparameters? Is there any thoretical or empirical guidance?

2. The proposed method is claimed to be able to find the global optimum of the aquisition functions in a scalable way. Is there a theoretical guarantee?  It seems that the performance of the posterior sampling procedure plays a key role for the follow-up root finding based global optimum searching procedure. Is the posterior sampling procedure scalable to higher dimensions? As far as we know, posterior sampling of a high dimensional parameter space is much more challenging issue to be resolved. If this procedure only yields low quality samples, how to guarantee that the proposed method here can work well?

---

> ### Author Response · Authors · 2024-11-25
> **Response to Reviewer kcrQ**
>
> Thank you for your thoughtful review and constructive feedback on our paper.
> We have carefully revised our paper, with some important new analytical and experimental results;
> please see the comment "Changes to the paper".
> Below are specific responses to the review.
>
> ### Summary
>
> > The common practice of BO is to utilize routine algorithms, such as ... Thompson sampling (TS), to solve the acquisition function optimization problem.
>
> It is worth clarifying that Thompson sampling (TS) should be seen as an acquisition policy,
> rather than a global optimization method.
> In particular, Gaussian process Thompson sampling (GP-TS)
> extends the classical TS for finite-armed bandits to continuous settings of BO.
> It takes a sample path of the GP posterior as a random acquisition function,
> whose global optimization is very challenging to be solved accurately.
> This is exactly the motivation of our paper.
>
> A common practice for GP-TS is to sample the GP posterior jointly on a set of
> a large number of points, which approximates the continuous domain,
> and the global optimum on the set is selected as the Thompson sample.
> But the true global optimum should be sought on the entire continuous domain.
>
> This is explained in Section "5 Related Works" paragraph "Posterior Sample-Based Acquisition Functions".
>
> > The basic idea is to select a subset of posterior samples as the starting points to conduct gradient-based root finding for searching local minima of the acquisition function.
>
> It is also worth clarifying that the starting points of TS-roots
> are not a subset of a large discrete sample of the GP posterior.
> Given a prior sample path $f(\mathbf{x})$,
> we obtain a posterior sample path $\widetilde{f}(\mathbf{x})$ via pathwise conditioning.
> The starting points of TS-roots consists of:
> (1) $n_{\text{e}}$ local minima of $f$ with the smallest $\widetilde{f}$ values;
> and (2) $n_{\text{x}}$ observed locations in $X$ with the smallest $\widetilde{f}$ values.
> The posterior sample path is not computed as a joint sample of the GP posterior
> on a large discrete set of points.
>
> This is explained in Section "3.1 TS-roots Algorithm".
> See also the new algorithmic box: "Algorithm 1 TS-roots".
>
> ### Weaknesses
>
> > The premise for that ... Unfortunately this is not proved or even discussed in the paper.
>
> We agree that it is an important question to study
> the effect of the quality of global optimization of acquisition functions
> on the quality of global optimization of the objective function in Bayesian optimization.
> We focus on the GP-TS acquisition policy, and our attempt to answer this question
> is given in our outer-loop results, see Figure 2.
> We used TS-RF and TS-DSRF as alternative methods for implementing GP-TS.
> In particular, TS-DSRF uses a similar decoupled representation of posterior sample paths
> as in TS-roots, but it uses random starting points for the gradient-based optimization,
> which is a common strategy for optimizing posterior sample paths as well as other acquisition
> functions in BO.
> The results on the 10D Levy and 16D Powell functions shows that the two alternatives
> give the worst results among all methods compared, while the results of TS-roots
> are either the best or comparable with the best.
>
> This means that GP-TS is a great acquisition policy, but its performance heavily depends on the
> global optimization method used. And we demonstrated that the TS-roots method can solve
> the inner-loop optimization problem accurately, which shows in the improved outer-loop performance.
>
> > Is it worth to develop more complex algorithms ... It's likely because it plays a more important role for BO's performance.
>
> We agree that building better surrogate models is important for BO,
> but this is not in conflict with the search for better global optimization algorithms
> for the inner loop. When the true objective function is complex, increasing the accuracy of
> the surrogate model also means increasing its complexity,
> which means that the corresponding acquisition functions are harder to optimize.
> We believe that finding a good global optimization algorithm is always an important task.
>
> In terms of the complexity of our proposed algorithm,
> we analyzed the computational cost of TS-roots in details in the new section:
> "C.5 Computational Complexity of TS-roots".
> The overall cost of TS-roots scales as $\mathcal{O}(d n)$,
> where $d$ is input dimension and $n$ is data size.
> Considering that $n$ is usually limited in Bayesian optimization due to the cost of data collection,
> the TS-roots algorithm should scale well to high dimensions.
> We also conducted new experiments to show that
> the number of starting points, which dictates the computational cost of TS-roots,
> can be set to small numbers (e.g., 10 or 25) without sacrificing the quality of the optimization.
> This means that TS-roots need not be computationally expensive.
> We intend to make our implementation of the TS-roots algorithm openly available,
> so that everyone can access the algorithm easily.

---

> ### Author Response · Authors · 2024-11-25
> **Additional Response to Reviewer kcrQ**
>
> ### Weaknesses (continued.)
>
> > As shown in Figure 2, for some cases, ... for solving challenging real-life problems.
>
> (1) TS-roots gives the most robust outer-loop performance for the test functions.
> That is, the results of TS-roots are at least on par with the alternative methods, if not better.
> As we explained in Section "6 Results" paragraph "Optimizing Benchmark Functions",
> "...TS-roots shows fast improvement in later stages.
> This is because GP-TS favors exploration, which delays rewards."
> The only case when the median performance of TS-roots is not the best
> is for the 16D Powell function (see Figure 2e), where EI performs slightly better.
> However, EI performs the worst in many other problems.
>
> (2) We used a comprehensive set of benchmark problem across a range of dimensions
> common in BO problems.
> Our focus was on synthetic benchmark problems to facilitate controlled comparisons,
> which are essential in evaluating the effectiveness of any BO algorithms.
> Additionally, the dimensions of the selected benchmark problems
> reflect the typical application range of BO algorithms in real-world scenarios (see Frazier, 2018).
>
> (3) We added a real-world problem in structure optimization,
> see Section "6 Results" paragraph "Optimizing Real-world Problem" and Appendix "G Ten-bar Truss".
> Figure 2(f) shows that TS-roots provides the best optimization result with rapid convergence,
> not only improving on althernative implementations of GP-TS (i.e., TS-RF and TS-DSRF),
> but also improving on althernative acquisition policies (i.e., EI and GP-UCB).
> We are also running experiments on other real-world problems
> to further establish the performance of TS-roots.
> The additional results will be included in the paper
> if we can get the results before the rebuttal period ends.
>
> ### Questions
>
> > There are some hyperparameters... Is there any thoretical or empirical guidance?
>
> Regarding the number of algorithm parameters ($n_o, n_e, n_x$),
> we thank Reviewer kcrQ for this critical question.
> We now added a new section: "Appendix D Minimum size of exploration and exploitation sets".
> Our new experiments show that it is safe to keep the size of exploration and exploitation sets
> $n_e$ and $n_x$ at small values (e.g., 10 or 25),
> and the size of the candidate exploration set $n_o$ at a medium value (e.g., 500),
> without sacrificing the quality of the optimization.
> Since the number of starting points dictates the computational cost of TS-roots,
> this means that TS-roots need not be computationally expensive.
>
> > The proposed method is claimed to ... Is there a theoretical guarantee?
>
> We only claim that the TS-roots algorithm can find the global optimum of posterior sample path
> with a high probability, depending on the values of the algorithm parameters $n_o, n_e, n_x$.
> In fact, these parameters can be set to small numbers while still retaining the optimization accuracy.
> This is discussed and empirically validated in the new section:
> "Appendix D Minimum size of exploration and exploitation sets".
>
> The computational complexity is analyzed in details in the new section:
> "C.5 Computational Complexity of TS-roots".
> The overall cost of TS-roots scales as $\mathcal{O}(d n)$,
> where $d$ is input dimension and $n$ is data size.
> This shows that the TS-roots algorithm effectively breaks the curse of dimensionality.
>
> > It seems that the performance of ... how to guarantee that the proposed method here can work well?
>
> The posterior sampling procedure via pathwise conditioning gives **exact** samples of GP posteriors,
> in the sense of equality in distribution. This observation is independent of the input dimension.
> The procedure scales to high dimensions, because its evaluation consists of evaluating the prior
> sample path once and evaluating the canonical basis functions at each data point once.
> This approach has recently been extended to the case of large data sets (Lin et al., 2023).
> These ensure that the procedure scales to both high dimensions and large data sets.
>
> This is now explained in Section "2 General Background" paragraph "Decoupled Representation of GP Posteriors".
>
> ---
>
> We hope our responses have addressed your concerns
> and that our revisions have significantly improved the paper.
> If so, we would greatly appreciate your consideration of increasing your scores.
> Please do not hesitate to let us know if you have any further questions or comments.

---

> > ### Comment · Reviewer_kcrQ · 2024-11-26
> >
> > Thanks authors for the feedback.
> >
> > I agree with you on your comments on Thompson sampling (TS) and GP-TS, especially your argument that "TS should be seen as an acquisition policy, rather than a global optimization method".  Thank you for clarifying that the starting points are composed of samples drawn from both the posterior and the prior.
> >
> > Given that the author's feedback has addressed some of my concerns (though my main concern, namely the proposed method's ability to handle complex real-world problems, still remains), I am willing to raise my score to 6.

---

> ### Author Response · Authors · 2024-12-04
> **Additional Response to Reviewer kcrQ**
>
> We thank the Reviewer kcrQ for raising their rating of our paper.
> Here we would like to address an additional point.
>
> In Weaknesses #1, a question was raised about the importance of global optimization of acquisition functions on the performance of BO algorithms. This is indeed an important but rarely answered question for the BO community, as the reviewer rightfully pointed out in Strengths #1, referring to this as "an uncommon topic of BO".
>
> A recent paper (Ament et al., 2023) shed light on this question for a different family of acquisition functions:
> expected improvement (EI) and its variants.
> (We thank the Reviewer MHUs for bringing this paper to our attention.)
> In that paper, the authors proposed to apply a log transform on the EI acquisition function (called LogEI) so that the function is no longer flat in most regions of the input space.
> They also used numerically stable implementations of the function to avoid numerically zero values.
> They showed that, with these tweaks, the EI acquisition policy is much easier to optimize
> and "surprisingly, are on par with or exceed the performance of recent state-of-the-art acquisition functions,
> highlighting the understated role of numerical optimization in the literature."
> _Their paper was accepted as a spotlight paper at NeurIPS 2023._
>
> Our paper addresses the difficulty in optimizing another type of acquisition functions, i.e., functions based on posterior sample paths.
> These functions, unlike most other acquisition functions, are not flat in most regions of the input space.
> This makes them even more difficult to optimize, and it does not admit a simple fix as those in LogEI.
> Yet, the main results of our paper show that the proposed TS-roots algorithm is able to make Gaussian process Thompson sampling (GP-TS) the most robust acquisition policy among the methods and problems we tested.
> As we explained in our earlier response, the outer-loop results of TS-roots are at least on par with the alternative methods, and oftentimes better.
> This observation is consistent across test functions and real-world problems.
> We hope that our paper makes a solid contribution to the BO community by
> advancing the topic of global optimization of acquisition functions
> and solving the unique challenges of optimizing posterior-sample-based acquisition functions.
>
> References:
> - Sebastian Ament, Samuel Daulton, David Eriksson, Maximilian Balandat, and Eytan Bakshy.
> Unexpected improvements to expected improvement for Bayesian optimization.
> In Advances in Neural Information Processing Systems, volume 36, pp. 20577–20612, 2023.
> https://openreview.net/forum?id=1vyAG6j9PE

---

### Official Review · Reviewer_D2as · 2024-11-02

**Soundness:** 3
**Presentation:** 3
**Contribution:** 3
**Rating:** 8
**Confidence:** 4

**Summary:**

The paper proposes a new methodology to optimize acquisition functions based on posterior sample paths. By judiciously selecting initial starting points, the proposed method makes a gradient-based multi-start strategy an appealing technique for optimizing the acquisition function. The effectiveness of the proposed methods has been demonstrated in multiple examples and comparisons with other existing methods are provided.

**Strengths:**

The paper delves into a problem that's important in Bayesian optimization but less understood, namely the effect or efficient strategy of optimizing the acquisition function. As the paper said, many (almost all) Bayesian optimization methods assume the exact inner optimization solution, which can be very difficult in practice. The authors provide a systematic way to come up with a starting point to facilitate existing multi-start gradient-based optimization strategies in acquisition functions. The effectiveness of this strategy and improving the existing sample path based acquisition function strategy looks impressive.

**Weaknesses:**

No major weakness. The paper is clearly written, except for the part where it describes the starting point for the inner acquisition optimization task. It seems that there are some inconsistencies between descriptions and notations between lines 142-161 on page 3. 1) I suggest authors revise this part to enhance the clarity of the method. 2) Furthermore, having an algorithmic box description would help readers gain the gist of the proposed method.

**Questions:**

1. The author describes the computational complexity of the algorithm in the appendix. It seems that finding local optimizers of the prior sample path is of O(m^2). It seems that this additional computational cost may be a bit of problematic in high(or even moderate) dimension, where you need more initial locations to optimize the inner acquisition function. Can you provide some computational efficiency comparison with other existing methods (based on LHS or uniform grid) to perform posterior sample path based acquisition function optimization?

2. Rather subtle question I have is, the robustness of the separability assumption on Kernel. It seems that if one is only interested in characterizing the optimum, utilizing the separable kernel instead of standard ones (like standard Matern Kernel) is sufficient for describing the geometry near the optima. Can you comment a bit on the necessity of the separability assumption for Kernel, besides the implementation or computaion aspect?

---

> ### Author Response · Authors · 2024-11-25
> **Response to Reviewer D2as**
>
> Thank you for your constructive comments on our paper and valuable suggestions!
> We have carefuly revised our paper, with some important new analytical and experimental results;
> please see the comment "Changes to the paper".
> Below are specific responses to the review.
>
> ### Weaknesses
>
> > The paper is clearly written, except for the part where it describes the starting point for the inner acquisition optimization task.
> > It seems that there are some inconsistencies between descriptions and notations between lines 142-161 on page 3.
> > 1) I suggest authors revise this part to enhance the clarity of the method.
> > 2) Furthermore, having an algorithmic box description would help readers gain the gist of the proposed method.
>
> We have carefully checked and slightly revised the part mentioned by Reviewer D2as
> to enhance the clarity of how we define the set of starting points for inner-loop optimization.
> In addition, we added an algorithmic box description
> to outline the procedure of the proposed TS-roots method,
> which we find helpful in formalizing our method.
> We hope that readers will find it informative as well.
>
> The set $S$ of starting points for inner-loop optimization is mathematically defined in Eq. (4)
> as the union of the exploration set $S_e$ found from the smallest local minima of the prior sample path
> and the exploitation set $S_x$ from the data.
> Detailed descriptions of all related algorithms are provided in Appendix C.
>
> ### Question
>
> > The author describes the computational complexity of the algorithm in the appendix.
> > It seems that finding local optimizers of the prior sample path is of O(m^2).
> > It seems that this additional computational cost may be a bit of problematic in high (or even moderate) dimension,
> > where you need more initial locations to optimize the inner acquisition function.
>
> We agree with Reviewer D2as that the computational complexity of TS-roots (Algorithm 1)
> increases with the input dimension.
>
> However, this increase does not arise from the complexity $O(m^2)$
> seemingly required for solving the eigenvalue problem of the colleague matrix,
> where $m$ is the degree of the Chebyshev polynomial used
> to approximate a univariate component function $f_i(x_i)$
> of the multivariate separable prior sample function $f(\mathbf(x))$.
> It does not arise from the number of starting points used
> for optimizing the posterior sample path either.
>
> Regarding the complexity $O(m^2)$, no eigenvalue problem in Line 2 of Algorithm 2 (`roots`)
> is solved of polynomial order greater than 100.
> If $m > 100$, then the interval is divided recursively into subintervals
> and a polynomial of appropriately lower degree is constructed on each subinterval.
> This is now explained in "C.2 Univariate Global Rootfinding".
>
> Regarding the number of starting points, we added a new section:
> "Appendix D Minimum size of exploration and exploitation sets".
> Our new experiments show that it is safe to keep the size of exploration and exploitation sets
> at small values (e.g., 10 or 25), without sacrificing the quality of optimization.
>
> The computational complexity of TS-roots is discussed in details in the new section:
> "C.5 Computational Complexity of TS-roots".
> In there, we discussed that the cost of the optimization procedure is dominated by
> `minsort`, evaluations of the posterior sample path $\widetilde{f}(\mathbf{x})$,
> and the gradient-based multistart optimization.
> The `minsort` algorithm calls `chebfun`, `roots`, and `maxk_sum`.
> The overall cost of TS-roots scales as $\mathcal{O}(d n)$,
> where $d$ is input dimension and $n$ is data size.
> This shows that the TS-roots algorithm effectively breaks the curse of dimensionality.
> Considering that $n$ is usually limited in Bayesian optimization due to the cost of data collection,
> the TS-roots algorithm should scale well to high dimensions.
>
> > Can you provide some computational efficiency comparison with other existing methods
> > (based on LHS or uniform grid) to perform posterior sample path based acquisition function optimization?
>
> As suggested by the reviewer,
> we have added new experimental results comparing TS-roots with uniform grid and the LHS:
> see "Appendix I Additional Results" paragraph "Comparison of Inner-loop Optimization Results".
> The comparisons show that TS-roots outperforms both uniform grid and LHS initialization schemes
> in terms of solution quality and computational cost.
> The performance of uniform grid and the LHS are qualitatively similar to random starting points.

---

> ### Author Response · Authors · 2024-11-26
> **Additional Response to Reviewer D2as**
>
> ### Question (continued.)
>
> > Rather subtle question I have is, the robustness of the separability assumption on Kernel.
> > It seems that if one is only interested in characterizing the optimum,
> > utilizing the separable kernel instead of standard ones (like standard Matern Kernel)
> > is sufficient for describing the geometry near the optima.
> > Can you comment a bit on the necessity of the separability assumption for Kernel,
> > besides the implementation or computaion aspect?
>
> We thank the reviewer for asking this excellent question!
>
> We consider a Bayesian optimization (BO) algorithm to consist of three key elements:
> (i) a prior probabilistic model of the objective function;
> (ii) an acquisition policy; and
> (iii) a global optimization algorithm to implement (ii).
>
> We believe that for a BO algorithm to robustly and efficiently minimize a given objective function,
> it only needs to approximate the function roughly in high-value regions
> while capturing the function in accurately low-value regions.
> Since a globally accurate approximation is unnecessary,
> this allows for flexibility in the surrogate modeling part of BO.
>
> The problem we are trying to solve in this work is the global optimization of posterior sample paths,
> which is very difficult due to its numerous local optima that grows exponentially in input dimension.
> Needless to say, some structure is necessary to allow for an efficient solution.
> We use the pathwise conditioning representation (Wilson et al., 2020) to relate
> a prior sample path, the prior covariance and the data set to a posterior sample path.
> One key to our TS-roots algorithm is to use the optima of the prior sample path
> as surrogates of the optima of the corresponding posterior sample path.
> (The observed locations are also used for this purpose, but they are readily available.)
>
> Separability of the Gaussian process prior is another key factor that enables our method.
> Many of our analytical results hinges on this assumption:
> (1) characterization of extrema (Appendix A.1);
> (2) counting numbers of extrema (Appendix A.2);
> (3) efficient filtering of the best extrema (Appendix B); and
> (4) efficient spectral sampling of the prior (Appendix C.1), to name a few.
> Excluding implementation or computational considerations, points (1) and (2) still remains.
> In particular, without point (1), we would have no idea about the locations of the local minima
> of a prior sample path.
> Other structures may also be imposed on the prior, such as additivity and its generalizations,
> but they tend to be more restrictive.
>
> We choose to impose separability because it is a very generic assumption to multivariate kernels.
> For example, the multivariate squared exponential kernel is separable.
> Although the standard multivariate Matern kernel is non-separable,
> we can construct a separable multivariate Matern kernel by multiplying univariate Matern kernels.
> Models of multivariate functions are usually constructed from univariate ones.
> In fact, the standard multivariate Matern kernel is generalized from the univariate Matern kernel
> by replacing the absolute value function with the Euclidean norm.
> But it is not clear what benefit this construction brings compared with the separable construction.
>
> Separable priors are not limiting,
> because as soon as any data point is collected, the posterior ceases to be separable in general.
> Of course, such priors do not account for any other _known_ structures in the objective function.
> If strong nonlinear dependencies exist between dimensions such as curved ridges or valleys,
> separable kernels may be less data efficient.
> But how one can improve the efficiency of a BO algorithm by injecting global information
> is a problem-specific and nontrivial question.
>
> ---
>
> We hope our responses have addressed your concerns
> and that our revisions have significantly improved the paper.
> If so, we would greatly appreciate your consideration of increasing your scores.
> Please do not hesitate to let us know if you have any further questions or comments.

---

### Official Review · Reviewer_MHUs · 2024-11-05

**Soundness:** 3
**Presentation:** 2
**Contribution:** 2
**Rating:** 6
**Confidence:** 3

**Summary:**

The paper proposes a method for optimizing the sample paths generated by GP posterior. The method, TS-root, globally optimizes the posterior samples via gradient-based multi-start optimization.
Specifically, TS-root can provide the gradient-based solver with a set of starting points accounting for exploration and exploitation, thereby finding the global optimum of posterior sample more efficiently. This starting point set S is discussed in section 3.1. Basically, S consists of exploration set S_e and exploitation set S_x. As S_x is a subset of observed data points satisfying eq (4), the number of points n_x can be small (hence easy to find, I think?). However, the set S_e is a n_e subset from the set S_0 of n_0 prior sample local optimum, and because n_0 can be larger, especially in high dimensions, finding S_e is non-trivial, which is discussed in Section 3.4 and 3.5. The main focus of the paper is to compute set S_e.

Another important component of TS-root is that it does not follow the conventional representation for GP posterior, and instead adopt the Decoupled form of GP posterior. This technique was presented in Wilson et al., 2020. The main idea of this decoupled form is to reduce to computational cost of conventional GP posterior, while maintaining similar predictive performance.

**Strengths:**

-	The paper addresses the problems of finding the global optimum of GP posterior samples efficiently, which is an important topic for many BO acquisition functions.
-	There are many empirical analyses for the proposed methods.
-	The method significantly reduces the CPU times to optimize GP-TS compared against genetic algorithm and random multi-start.
-	The method can be a robust choice for general problem – TS-root performance is the most robust among other baselines, despite only being on par with the best baselines in each problem.

**Weaknesses:**

-	The contribution is fair, as the main contribution seems to be computing the starting points for gradient-based optimization process. The decoupled representation for GP posterior was proposed by previous works.
-	The idea seems to not very effective in improving MES acquisition function. Fig. 5 shows that TS-Random Fourier and TS-root are statistically similar.
-	No real-world benchmark problems.

**Questions:**

-	There are also other works addressing the issue on mostly-flat acquisition functions (stated by authors in line 49). One of the notable approaches is to compute log of acquisition values, which has been shown to mitigate the flatness problem. One example is a recent work [1], which combines previous similar attempts into a systematic solution, the logEI acquisition function. Can the authors comment on this log approach? For example, how about computing log acquisition values + random multi-start for gradient solver?

[1] Ament, Sebastian, et al. "Unexpected improvements to expected improvement for bayesian optimization." Advances in Neural Information Processing Systems 36 (2023): 20577-20612.

---

> ### Author Response · Authors · 2024-11-25
> **Response to Reviewer MHUs**
>
> Thank you for your thorough review and valuable feedback on our work.
> We have carefully revised our paper, with some important new analytical and experimental results;
> please see the comment "Changes to the paper".
> Below are specific responses to the review.
>
> ### Summary
>
> > Another important component of TS-root is that it does not follow the conventional representation for GP posterior, and instead adopt the Decoupled form of GP posterior. This technique was presented in Wilson et al., 2020. ...
>
> It is worth clarifying the importance of the pathwise conditioning representation of GP posteriors (Wilson et al., 2020; 2021),
> which is a key building block of this work.
> This representation is exact, efficient, preserves differentiability of the prior, and has its origins in the classical GP literature.
> We have thus added the following to Section 2 paragraph "Decoupled Representation of GP Posteriors":
>
> "This representation has its roots in Matheron’s update rule that transforms a joint distribution of Gaussian variables into a conditional one (see e.g., Hoffman & Ribak (1991)). This formula is exact, in that $\overset{\text{d}}{=}$ denotes equality in distribution, and it preserves the differentiability of the prior sample. It is also computationally efficient for posterior sampling: the cost is independent of the input dimension $d$, linear in the data size $n$ at evaluation time, and the weight vector for $\boldsymbol{\kappa}_{\cdot,n}(\mathbf{x})$ can be solved accurately using an iterative algorithm that scales linearly with $n$ (Lin et al., 2023)."
>
> ### Weaknesses
>
> > The contribution is fair, as the main contribution seems to be computing the starting points for gradient-based optimization process. The decoupled representation for GP posterior was proposed by previous works.
>
> We agree that the decoupled representation of GP posterior is not our contribution,
> which we made clear by putting it in Section "2 General Background".
>
> The effect of the selected starting points is non-trivial, considering that random starting points
> cannot find the global optimum of posterior sample paths accurately,
> especially in higher dimensions (see Figure 4).
> This inaccuracy in the inner loop leads to poor performance in the outer loop
> (see Figure 2 (c, e) TS-DSRF and TS-RF curves).
> As suggested by another reviewer,
> we also added new experimental results of gradient-based multistart optimization
> with other initialization schemes (i.e., uniform grid and Latin hypercube sampling),
> see "Appendix I Additional Results" paragraph "Comparison of Inner-loop Optimization Results".
> The comparisons show that TS-roots outperforms both uniform grid and LHS
> in terms of solution quality and computational cost.
>
> _To find good starting points is to solve the optimization problem approximately._
> In fact, as evidenced in our new results
> (see Appendix "D Minimum Size of Exploration and Exploitation Sets" and Figure 7),
> using the first point in $S_e$ and the first point in $S_x$ (sorted by posterior sample function values)---**only two points**---we
> can discover the global optimum **most of the time**.
> Furthermore, the new results verified that the number of starting points ($n_e$ and $n_x$)
> can be set to small numbers (e.g. 10 or 25), independent of the input dimension and the BO iteration,
> without sacrificing optimization accuracy.
> This means that TS-roots is an efficient and scalable approach
> for the global optimization of posterior samples.
>
> We believe this work will significantly improve the efficiency and performance
> of Gaussian process Thompson sampling, and allow it scale to high dimensions and large datasets.
>
> > The idea seems to not very effective in improving MES acquisition function.
>
> We agree that the results of TS-roots-based MES did not significantly outperform
> those based on TS random Fourier features.
> This can be attributed to two key factors that influence the performance of the adopted MES-R method:
> (1) the quality of samples generated from the conditional distribution of minimum values
> to construct the MES acquisition function,
> and (2) how good the optimization of the MES acquisition function is.
> While the number of samples to approximate the conditional distribution of objective function values
> is limited, the second factor plays a more significant role in determining the optimization results.
> In our MES experiment, we focused on addressing the first issue via our TS-roots algorithm,
> leaving the second issue unaddressed.
> Thus, even modest improvements in the optimization result indicate that
> our TS-roots algorithm provides a better set of posterior function values
> than TS random Fourier features.
>
> To clarify this point, we have added the following sentence to
> Section "6 Results" paragraph "TS-roots to Information-Theoretic Acquisition Functions":
>
> "Note that the inner-loop optimization of MES, which strongly influences the optimization results,
> is not addressed by TS-roots."

---

> ### Author Response · Authors · 2024-11-25
> **Additional Response to Reviewer MHUs**
>
> ### Weaknesses (continued.)
>
> > No real-world benchmark problems.
>
> Our focus was on synthetic benchmark problems to facilitate controlled comparisons,
> which are essential in evaluating the effectiveness of any Bayesian optimization algorithms.
> Additionally, the dimensions of the selected benchmark problems reflect the typical application range
> of Bayesian optimization algorithms in real-world scenarios (Frazier, 2018).
>
> We now added a real-world problem in structure optimization,
> see Section "6 Results" paragraph "Optimizing Real-world Problem" and Appendix "G Ten-bar Truss".
> Figure 2(f) shows that TS-roots provides the best optimization result with rapid convergence,
> not only improving on alternative implementations of GP-TS (i.e., TS-RF and TS-DSRF),
> but also improving on alternative acquisition policies (i.e., EI and GP-UCB).
>
> We are also running experiments on other real-world problems
> to further establish the performance of TS-roots.
> The additional results will be included in the paper
> if we can get the results before the rebuttal period ends.
>
> ### Questions
>
> > There are also other works addressing the issue on mostly-flat acquisition functions (stated by authors in line 49). ... For example, how about computing log acquisition values + random multi-start for gradient solver?
>
> Regarding the LogEI approach (and its extensions),
> this is a good solution to the pathology of flat acquisition surface
> over large regions of the input variable space.
> Such flat regions can lead to vanishing gradient issues that make Bayesian optimization
> extremely sensitive to the initial conditions of acquisition function maximization.
> Using log transformation, the modified acquisition functions are less dependent on
> heuristic initialization strategies, which justifies the selection of random starting points.
> The log approach works for acquisition functions prone to the flat surface problem,
> including the EI-based family, and potentially PI and some information-based acquisition functions.
>
> However, unlike most acquisition functions, posterior sample paths do not exhibit flat surface issues
> but tend to have many local minima (as illustrated in Fig. 1).
> Thus, optimizing these acquisition functions requires careful selection of starting points.
> We have added the discussion on the LogEI approach to
> Section "5 Related Works" paragraph "Optimization of Acquisition Functions":
>
> "The log reformulation approach is a good solution to the numerical pathology of flat acquisition surface over large regions of the input variable space (Ament et al., 2023). While this approach works for acquisition functions prone to the flat surface issue such as the family of EI-based acquisition functions, its performance has yet to be evaluated for acquisition functions with many local minima like those based on posterior samples."
>
> ---
>
> We hope our responses have addressed your concerns
> and that our revisions have significantly improved the paper.
> If so, we would greatly appreciate your consideration of increasing your scores.
> Please do not hesitate to let us know if you have any further questions or comments!

---

> > ### Comment · Reviewer_MHUs · 2024-11-29
> >
> > Thank you for the authors' response. The response has addressed my main concerns, so I will increase my score.

---

### Author Response · Authors · 2024-11-25
**Changes to the paper**

After considering the initial reviewer comments,
we have carefully made the following changes to the paper:

- New algorithmic box for the proposed `TS-roots` method (Algorithm 1).
- Revised `minsort` algorithm allowing _exact_ subsetting of prior local minima (Algorithm 4).
- New analysis on the computational complexity of `TS-roots` (Appendix C.5).
- New analytical result on the number of local minima of a separable function (Appendix A.2)
- New empirical results on the effect of set sizes $n_o, n_e, n_x$ on optimization accuracy
  (Appendix D).
- New empirical result on the out-loop performance of TS-roots in real-world problem
  (Section 6 paragraph "Optimizing Real-world Problem"; and Appendix G)
- New empirical result on gradient-based multistart optimization using
  two other initialization schemes: uniform grid and Latin hypercube sampling.
  (Appendix I paragraph "Comparison of Inner-loop Optimization Results")
- Further explanation of the pathwise conditioning representation of GP posterior (Wilson et al., 2020; 2021), which is an important building block of our proposed TS-roots method.
- New discussion on the log transform of acquisition function
  (Section 5 paragraph "Optimization of Acquisition Functions")
- Clarifications on: contributions of the paper; the `TS-roots` algorithm;
  improvement in the performance of MES; spectral representation of a separable GP prior;
  computational complexity of the `roots` algorithm;
- Move the description of Sample-Average Posterior Function to a new section (Section 4).

Changed text are highlighted in blue in the revised manuscript PDF.

---

### Meta-Review · Area_Chair_rTv8 · 2024-12-19

**Metareview:**

This study proposes a novel method for the global optimization of acquisition functions by optimizing sample paths generated from the posterior distribution of Gaussian processes (GPs). The key feature of this method is its utilization of the decoupled representation of the GP posterior. By leveraging this decoupled framework, the proposed method provides a systematic algorithm based on gradient optimization that promotes the use of multiple initial points.　A potential weakness is the assumption of separability. However, as the authors point out, this assumption is satisfied by the squared exponential kernel and can be met for the Matérn kernel with appropriate adjustments, making it unlikely to pose a significant practical limitation.
All reviewers have acknowledged the significant contributions of this paper, and numerical experiments further demonstrate its utility. Therefore, I believe this work is valuable to the community and recommend acceptance.

**Additional Comments On Reviewer Discussion:**

Reviewer D2as raised concerns about the assumption of separability, but the authors explained that this does not pose a practical limitation. Reviewer MHUs pointed out the lack of evaluation using benchmark datasets, but this concern was addressed through the addition of new experiments. Reviewer D2as also raised concerns about computational costs, which were resolved through additional discussions provided by the authors. In summary, the authors have adequately addressed all concerns raised by the reviewers, and I believe this study is ready for publication.

---

### Decision · Program_Chairs · 2025-01-22

Accept (Poster)